# SNM1A is crucial for efficient repair of complex DNA breaks in human cells

Lonnie P. Swift [1], B. Christoffer Lagerholm[2,6], Lucy R. Henderson [1], Malitha Ratnaweera [1], Hannah T. Baddock[1,7], Blanka Sengerova [1,8], Sook Lee[1], Abimael Cruz-Migoni [1], Dominic Waithe[2], Christian Renz [3], Helle D. Ulrich [3], Joseph A. Newman [4], Christopher J. Schofield [5] & Peter J. McHugh [1] ✉

DNA double-strand breaks (DSBs), such as those produced by radiation and radiomimetics, are amongst the most toxic forms of cellular damage, in part because they involve extensive oxidative modifications at the break termini. Prior to completion of DSB repair, the chemically modified termini must be removed. Various DNA processing enzymes have been implicated in the processing of these dirty ends, but molecular knowledge of this process is limited. Here, we demonstrate a role for the metallo-β-lactamase fold 5′−3′ exonuclease SNM1A in this vital process. Cells disrupted for SNM1A manifest increased sensitivity to radiation and radiomimetic agents and show defects in DSB damage repair. SNM1A is recruited and is retained at the sites of DSB damage via the concerted action of its three highly conserved PBZ, PIP box and UBZ interaction domains, which mediate interactions with poly-ADP-ribose chains, PCNA and the ubiquitinated form of PCNA, respectively. SNM1A can resect DNA containing oxidative lesions induced by radiation damage at break termini. The combined results reveal a crucial role for SNM1A to digest chemically modified DNA during the repair of DSBs and imply that the catalytic domain of SNM1A is an attractive target for potentiation of radiotherapy.

DNA double-strand breaks (DSBs) are one of the most cytotoxic forms of damage, with even a small number of unrepaired breaks being potentially lethal[1]. In human cells, the majority of DSBs are resolved by one of two pathways, that is non-homologous end-joining (active from G1- through to the G2-phase of the cell cycle) or homologous recombination (which is activated as cells traverse S-phase)[2]. Before completion of DSB repair, any chemically modified nucleotides or aberrant structures must be removed from the break-ends. This process is especially important for DSBs induced by ionising radiation (IR) or radiomimetic drugs, including bleomycin and related agents, associated with extensive, mostly oxidative, DNA modifications at the break termini[3,4]. Several factors have been implicated in processing such 'dirty end' ends, including tyrosyl DNA phosphodiesterase (TDP1)[5], polynucleotide kinase (PNK)[6], aprataxin[7–10] and Artemis/SNM1C (discussed below)[11,12]. TDP1 can remove the 3′-phosphoglycolate ends that constitute approximately 10% of the termini produced by IR[13]. PNK catalyses the removal of 3′-phosphate groups and the addition of phosphates to 5′-hydroxyl moieties in preparation for end

[1]Department of Oncology, MRC-Weatherall Institute of Molecular Medicine, University of Oxford, John Radcliffe Hospital, Oxford, United Kingdom. [2]Wolfson Imaging Centre, MRC-Weatherall Institute of Molecular Medicine, University of Oxford, John Radcliffe Hospital, Oxford, United Kingdom. [3]Institute of Molecular Biology gGmbH (IMB), Mainz, Germany. [4]Centre for Medicines Discovery, University of Oxford, Oxford, United Kingdom. [5]Chemistry Research Laboratory, Department of Chemistry and the Ineos Oxford Institute for Antimicrobial Research, University of Oxford, Oxford, United Kingdom. [6]Present address: Cell Imaging and Cytometry Core, Turku Bioscience Centre, University of Turku and Åbo Akademi, ku, Finland. [7]Present address: Calico Life Sciences, South San Francisco, CA, USA. [8]Present address: Institute of Organic Chemistry and Biochemistry of the Czech Academy of Sciences, Prague, Czech Republic. ✉e-mail: peter.mchugh@imm.ox.ac.uk

ligation[6], modifications associated with oxidation reactions following IR. Aprataxin catalyses deadenylation releasing DNA Ligase IV during abortive ligation reactions during NHEJ, removing the associated AMP group[14].

The SNM1A 5′−3′ exonuclease encoded by the *DCLRE1A* gene is a member of a family of metallo-β-lactamase (MBL) fold DNA nucleases conserved, at least, from yeasts to humans[15]. The family is characterised by the presence of a β-CASP (CPSF-Artemis-SNM1A-Pso2) domain that together with the MBL domain forms the nuclease active site[16]. Human cells have three MBL-β-CASP DNA nuclease paralogues: SNM1A, SNM1B (also known as Apollo) and SNM1C (Artemis)[16].

SNM1A has a key role in DNA interstrand crosslink repair (ICL repair), where a role in the replication-coupled pathway of ICL repair in mammalian cells is mediated through interaction of SNM1A with PCNA via its PCNA interacting protein (PIP) motif as well as interaction via an N-terminally located ubiquitin-binding zinc finger (UBZ) with the monoubiquitinated form of PCNA[15,17,18]. Very recently, SNM1A has been shown to be important in mediating DNA damage tolerance associated with break-induced replication and recombination at telomeres maintained by the alternative lengthening (ALT) pathway[19]. While the role of SNM1B/Apollo in DNA repair has not been comprehensively elucidated, although it has been implicated in DSB and ICL repair, it has a well-defined role in the processing of newly synthesised telomere leading strands to maintain the structure at telomere termini[20].

SNM1A and SNM1B/Apollo are both 5′−3′ exonucleases[21]. SNM1C/ Artemis has a distinct catalytic profile, with limited 5′−3′ exonuclease activity. On association with DNA-PKcs (the catalytic subunit of the NHEJ factor DNA-PK), the 5′−3′ endonuclease activity of SNM1C/Artemis acquires the capacity to open DNA hairpins and to remove overhangs and unpaired regions at damaged/aberrant DNA termini[12,22]. SNM1C/Artemis plays a key role in V(D)J recombination, by removing RAG-generated hairpin intermediates and in processing of DNA breaks bearing modified termini, including oxidised nucleotides induced by IR as part of the NHEJ pathway

Here we report that cell lines disrupted for SNM1A by genome-editing display the anticipated increased sensitivity to DNA crosslinking agents. Unexpectedly, screening more generally for DNA damage sensitivity revealed the sensitivity of SNM1A disrupted cells to radiation and radiomimetic agents. This unanticipated observation led us to comprehensively characterise a DSB repair role for SNM1A. We demonstrate that several ligand associations act to recruit SNM1A to DSBs and that its DSB repair role likely reflects the exceptional capacity of SNM1A to hydrolyse DNA break substrates with chemical and structural modifications.

## Results

### SNM1A⁻ cells show sensitivity to radiometric damage

To investigate the phenotype of human cells disrupted for SNM1A we utilised both zinc-finger Nuclease (ZFN)[23] and CRISPR-Cas9[24] gene editing to generate SNM1A⁻ cells. Initially, we employed ZFN constructs targeting the first exon of SNM1A. A putative U2OS SNM1A disrupted clone was identified, containing a 22-nucleotide deletion in exon 1, producing a frameshift and premature stop codon distal from the deletion site (Supplementary Fig. 1); SNM1A protein was not detected by immunoblotting in these cells (Fig. 1A).

To examine the DNA damage response defects in cells lacking SNM1A, we treated them with a broad range of genotoxic agents and determined their survival relative to the parental U2OS cells in clonogenic survival assays. As anticipated from previous work[17], SNM1A⁻ cells were more sensitive to the crosslinking agents cisplatin and SJG-136, supporting an important role for SNM1A in removing ICLs (Fig. 1A, B). Introduction of EGFP-SNM1A into the SNM1A⁻ cell line restored the cisplatin sensitivity of the SNM1A⁻ cells to near wildtype levels (Fig. 1A). Sensitivity to UVC irradiation was observed, possibly as a consequence of the rare ICLs induced by this form of damage (Supplementary

Fig. 2A). An absence of increased sensitivity was observed for a range of other genotoxins in SNM1A⁻ cells, including formaldehyde (HCHO), hydrogen peroxide (H₂O₂; which primarily produces oxidised bases and single-strand breaks) and the topoisomerase 1 inhibitor camptothecin, relative to the parental U2OS cells (Supplementary Fig. 2B, C, D). Strikingly, with the radiomimetic drug Zeocin, substantially increased sensitisation was observed for the SNM1A⁻ cells. This sensitivity was suppressed by stable complementation with EGFP-SNM1A (Fig. 1C). Since the principal genotoxic lesion induced by Zeocin is DSBs associated with chemically modified termini[25], we assessed the sensitivity of SNM1A⁻ cells to ionising radiation (IR) which induces related damage (Fig. 1D). Sensitivity to IR was observed, although the overall increase in sensitisation was lower than for Zeocin.

Together the above-described results suggest a role for SNM1A in the repair of the major cytotoxic lesions induced by Zeocin and IR; DSBs associated with chemically modified termini. To strengthen this proposal, we used CRISPR-Cas9 to target a sequence in the first exon of SNM1A in 293FT cells (Supplementary Fig. 1) and identified clones with frameshift mutations resulting in a distal introduction of a premature stop codon; these cells do not express SNM1A as determined by immunoblot (Fig. 1A). As with the U2OS SNM1A⁻ cell results, the 293FT cells manifest clearly increased sensitivity to Zeocin treatment (Fig. 1E), which is complemented by stable expression of an EGFP-SNM1A construct.

### SNM1A⁻ cells are defective in the repair of radiation and radiomimetic damage

As SNM1A is a 5′-3′ repair exonuclease, a plausible explanation for the sensitivity of SNM1A cells to Zeocin and IR is defective processing of the toxic DSBs these agents induce. To determine whether SNM1A⁻ cells exhibit the hallmarks of a DSB repair defect, we initially employed pulsed-field gel electrophoresis (PFGE) with the aim of directly detecting Zeocin and IR-induced DSBs and monitoring their resolution. However, in the cell lines employed, we found that the doses of Zeocin or IR required to permit the detection of DSBs were too high for repair to be monitored by PFGE in parental U2OS or 293FT cells. Consequently, we moved to monitoring the dynamics of γH2AX and 53BP1 foci, which are markers of DSB induction[26,27]. Following treatment of U2OS cells, with Zeocin or IR, γH2AX and 53BP1 levels peaked within 2 h, and were largely resolved within 24 h (Fig. 1F–H, representative images for Zeocin foci shown in Fig. 1I, with a wider field of cells shown in Supplementary Fig. 3). However, in SNM1A⁻ cells treated with Zeocin, a substantial fraction of γH2AX and 53BP1 foci persisted after 24 h (Fig. 1F, G), and comparable results were obtained for 293FT cells and their cognate SNM1A⁻ derivatives (Supplementary Fig. 4). Employing IR as the DSB-inducing agent also led to a marked delay in γH2AX foci resolution in SNM1A cells (Fig. 1H). Analysis of cell cycle progression following Zeocin treatment (0.1 mg/mL) demonstrated that parental (U2OS) cells only transiently slowed and arrested, whereas the SNM1A⁻ derivatives accumulated in the late S/G2 phase of the cell cycle for up to 48 h, characteristic of cells accumulating unrepaired DSBs (Fig. 1J, Supplementary Fig. 5A–D). Similarly, 293FT cells disrupted for SNM1A exhibited a greater proportion of cells accumulating in G2/M 24 h following Zeocin treatment (Supplementary Fig. 5E). Moreover, treatment of cells with IR (5 Gy) also led to the formation of EGFP-SNM1A foci many of which are proximal to γH2AX foci, as did treatment with cisplatin which is established to induced replication-associated SNM1A foci (Supplementary Fig. 6). Together, these observations demonstrate that SNM1A-deficient cells are proficient in signalling the presence of damage, but impaired in their repair-response to complex DSBs.

We then examined whether SNM1A is recruited to sites of Zeocin-induced DSBs, by employing cells that stably express a functional N-terminally EGFP-tagged form of SNM1A (Fig. 1A). By employing 53BP1 as a marker for DSB induction and localisation, we observed that

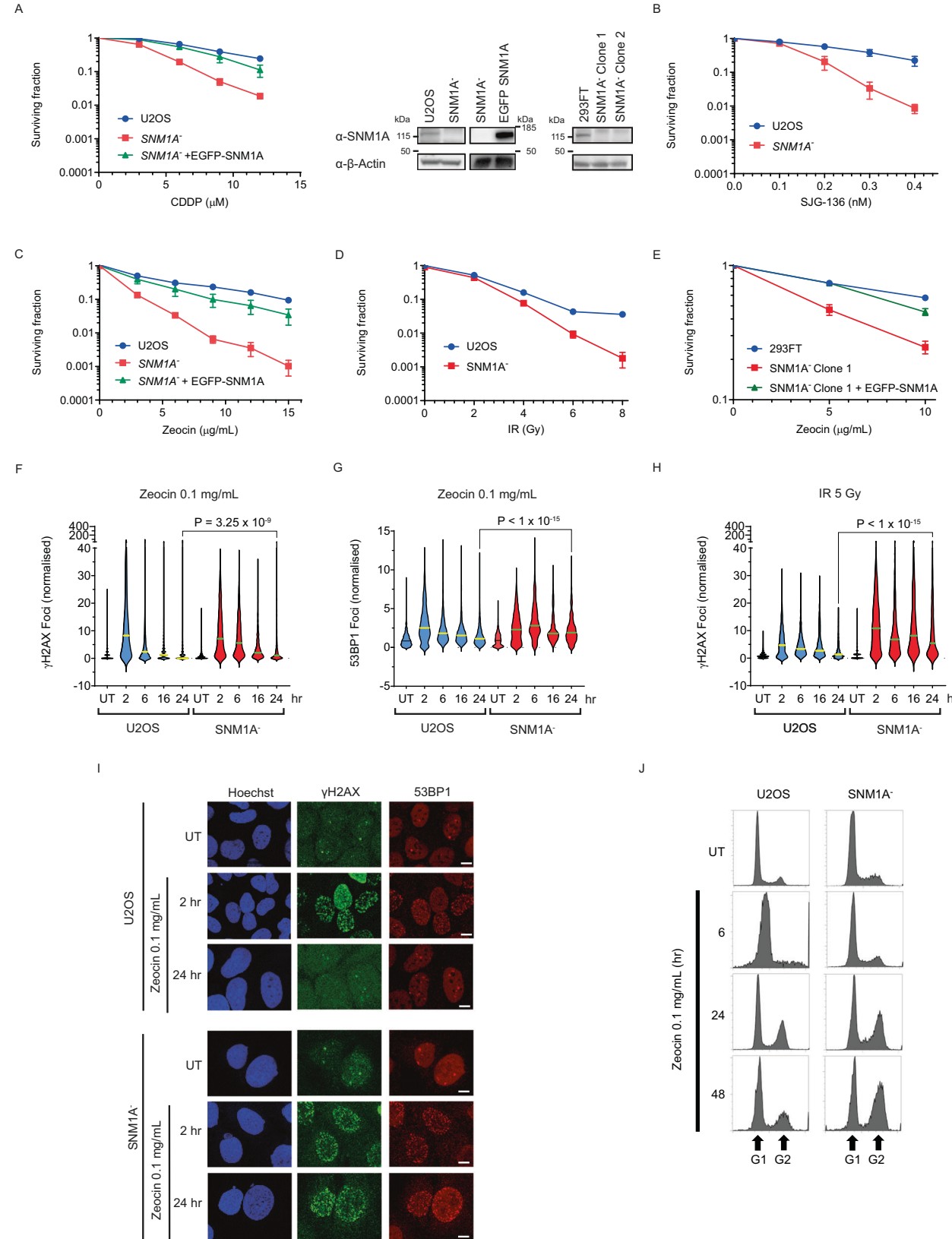

Zeocin treatment causes the majority of 53BP1 foci to localise in the proximity with EGFP-SNM1A foci, where SNM1A frequently localises adjacent to these 53BP1 foci and in some cases the signal overlaps (Fig. 2A, B and wider field view of multiple cells is shown in Supplementary Fig. 7). The observation that SNM1A localises to the vicinity of complex DSBs further supports the proposal that SNM1A plays an

important role in responding to complex DSBs. Moreover, EGFP-SNM1A localises in the proximity of γH2AX foci following treatment with IR, as well as cisplatin where a role in processing damage is established (Supplementary Fig. 6).

Interestingly, when reporter assays were used to investigate overt defects in HR or NHEJ induced by the 'clean' DSBs produced by the

**Fig. 1 | SNM1A⁻ cells show sensitivity to radiometric damage. A** SNM1A⁻ (U2OS) cells are sensitive to Cisplatin (CDDP) treatment (4 h) in a clonogenic survival assay. Sensitivity is suppressed in cells stably expressing an EGFP-SNM1A (left-hand panel, *n* = 3 biological replicates performed in duplicate, error = SEM). SNM1A protein is not detected in the SNM1A⁻ cells by immunoblot (right-hand panel), compared to wildtype U2OS cells, and the EGFP-SNM1A band is evident in the complemented cell line as shown in the inset panels. **B** SNM1A⁻ cells are sensitive to continuous treatments of the DNA crosslinking agent SJG-136 (*n* = 4 biological repeats, error = SEM). **C**. SNM1A⁻ cells show sensitivity to the radiomimetic Zeocin (continuous) and sensitivity is rescued by expressing EGFP-SNM1A (*n* = 3 biological replicates performed in duplicate, error = SEM). **D** SNM1A⁻ cells are sensitive to ionising radiation (IR, *n* = 4 biological repeats, error = SEM). **E** Zeocin sensitivity is observed in continuously treated SNM1A⁻ 293FT cells, SNM1A protein is not detected in these cell lines by immunoblot analysis (as shown in right-hand panel (**A**), *n* = 2 biological repeats, error = SD). Zeocin treatment (0.1 mg/mL, 2 h [hr]) induces persistent γH2AX (**F**) and 53BP1 (**G**) foci in SNM1A⁻ cells compared to untreated control (UT). **H** shows a similar response of γH2AX following IR treatment (foci data in (**F–H**) *n* = 3 biological repeats from left to right, (**F**) contains the data for 2370, 2307, 2105, 2138, 1764, 1098, 1619, 1028, 1358, 913 cells. **G** 1694, 1324, 1610, 810, 1786, 849, 820, 826, 548, 618 cells. **H** 2007, 1816, 1565, 1246, 1624, 2313, 1808, 1706, 1141, 1266 cells). *P* values calculated with the Kruskal-Wallis test using post-hoc Dunn's multiple comparison test. Panel (**I**) representative foci images of the data from (**F**) **and** (**G**) (Scale bars = 10 μm). **J** Cell cycle distribution by DNA content determined by flow cytometry following Zeocin treatment (0.1 mg/mL, continuous) over 48 h in SNM1A⁻ and the parental U2OS cells. Representative images of 3 biological repeats. Source data are provided as a Source Data file.

I-SceI endonuclease, no severe repair defects were observed in SNM1A⁻ cells (Supplementary Fig. 8A, B). By contrast, and as anticipated, cells disrupted for XRCC4 or cells depleted for BRCA2, acted as positive controls for defects in NHEJ or HR respectively[28,29]. A slightly elevated level of HR was observed for SNM1A⁻ cells, implying a minor, but not essential, role in DSB repair pathway utilisation at clean breaks. To further explore this point we determined whether SNM1A is recruited to the sites of 'clean' (i.e., not associated with chemically modified termini) DSBs using cells expressing a 4-hydroxy tamoxifen (4OHT)-inducible form of the endonuclease *AsiSI* which can cut multiple sites (at a greater than 1Mbp distance from each other) in the human genome[30]. Here, we observed a degree of recruitment of EGFP-SNM1A to the nuclear sites, where a subset of these co-localised with γH2AX (used to define the DSB sites), implying that SNM1A is recruited to a subset of clean DSBs at some stage in their processing (Fig. 2C, D). However, the lack of a strong HR defect suggests that SNM1A does not play a key role in the canonical pathways of end-resection that are required to generate the requisite 3′-overhangs for HR. Therefore, only DNA DSBs associated with complex, chemically altered termini, such as those induced by radiation and radiomimetics, appear to be strongly dependent on SNM1A for repair.

## The PBZ, UBZ and PIP box act in concert to recruit SNM1A to complex DNA breaks

The N-terminus of SNM1A is predicted to contain three highly-conserved motifs: a putative poly-ADP-ribose (PAR)-binding zinc finger (PBZ)[15,31], a ubiquitin-binding zinc finger (a UBZ4 motif, hereafter UBZ)[15,32] which is involved in the localisation of SNM1A to ICLs in S-phase through mediating interaction with ubiquitinated PCNA[18], and a PIP (PCNA interacting peptide) box[15,33] (Fig. 3A; sequence alignments are in Supplementary Fig. 9). Evidence for the direct interaction of these motifs with any of their predicted ligands has not been reported. As structural insight into the non-catalytic N-terminal region of SNM1A is currently lacking, likely due to extensive overall disorder, we predicted the architectures of the UBZ and PBZ domains, and PIP box using the AlphaFold platform[34]. The predicted SNM1A UBZ domain structure is similar to that of the UBZ4 domain of RAD18[35], except for a change in the conformation of the secondary structure of the Zn²⁺ coordinating residues of its C3H1-type zinc finger (Fig. 3B, left-hand panel); the predicted SNM1A PBZ domain structure closely matched the experimental PBZ structure of APLF[36] with the PBZ residues C155 and C161 being predicted to coordinate Zn²⁺ (Fig. 3B, middle panel); the PIP box prediction for SNM1A agreed strongly with the experimental (high affinity) PIP box structure of P21/WAF1[37], with critical residues Q556, I559, Y562 and F563 lying close to the expected grooves in PCNA (Fig. 3B, right-hand panel).

To explore these ligand interactions experimentally, we designed and produced a series of N-terminally GST-tagged peptides spanning the PBZ-plus-UBZ (residues 114–181) or PIP box (residues 547–575) motifs of SNM1A (Supplementary Fig. 9) and tested their capacity to interact with purified PCNA, purified lysine 164 ubiquitinated PCNA (PCNAᵘᵇ) or PAR chains. The results revealed that wildtype PIP box peptides (GST-PIP) efficiently bind PCNA and PCNAᵘᵇ, respectively, whereas a peptide containing substitutions at conserved residues in the relevant binding motifs, GST-PIPᵐᵘᵗ (Y562A, F563A double substitution) did not pull-down either PCNA or PCNAᵘᵇ (Fig. 3C, left-hand panel). While a GST peptide spanning the PBZ and UBZ motifs (GST-PBZ-UBZ) did not interact with native PCNA, PCNAᵘᵇ was pulled down by this peptide, indicating the UBZ motif-containing peptide can interact with PCNAᵘᵇ (Fig. 3C right-hand panel). A peptide harbouring structurally predicted ligand binding mutation in the UBZ, a GST-PBZ-UBZᵐᵘᵗ peptide (C125F substitution), did not interact with PCNAᵘᵇ (or PCNA). This is consistent with the mutated UBZ residue mediating interaction with PCNAᵘᵇ as described above. Likewise, GST-PBZ-UBZ peptides were able to bind PAR chains in a dot-blot analysis, whereas GST-PBZᵐᵘᵗ-UBZ (C155A and C161A double substitution) were not (Fig. 3D, upper panels). Purified APLF, an established PBZ-containing protein[31], acted as a positive control (Fig. 3D, lower panels).

Having confirmed that the PBZ, UPZ and PIP box mediate the predicted interactions in vitro, we investigated the potential role of each of these in the recruitment of SNM1A to complex DSBs. Cells expressing mutant forms of EGFP-SNM1A that ablate ligand interactions with PAR chains, ubiquitin and the PIP box, (EGFP-SNM1ᴾᴮᶻ, double C155A, C161A substitution; EGFP-SNM1Aᵁᴮᶻ, C125F substitution; EGFP-SNM1Aᴾᴵᴾ Y562A, C161A double substitution, respectively) were expressed in U2OS cells lacking endogenous SNM1A and the induction of SNM1A and 53PB1 foci formation following Zeocin treatment was monitored (Fig. 3E, F) As before, wildtype EGFP-SNM1A protein was proximal with 53BP1 following Zeocin treatment; this focus formation was independent of SNM1A catalysis, since cells expressing a SNM1A containing nuclease-inactivating mutation[17] (harbouring a D736A substitution, here denoted SNM1Aᴺᴵ) was also efficiently recruited to the sites of Zeocin DSBs. Examination of the SNM1Aᴾᴮᶻ, SNM1Aᵁᴮᶻ and SNM1Aᴾᴵᴾ variants demonstrated a trend towards reduction in recruitment to Zeocin-induced foci that were proximal with 53BP1. The post-treatment increase in SNM1A foci was not statistically significant for the PBZ mutant, and exhibited reduced significance for the UBZ and PIP mutant forms, compared with the wildtype and nuclease inactive forms of SNM1A (Fig. 3G) which both exhibited significant post-treatment increase in foci. Interestingly, when we evaluated the spatial relationship of these Zeocin-induced EGFP-SNM1A foci relative to 53BP1 and γH2AX foci, there was a clear trend that SNM1Aᴾᴵᴾ cells exhibited a reduced average number of foci proximal or overlapping with both DSBs markers (Supplementary Fig. 10), implying a role for the SNM1A PIP box in the localisation of SNM1A to complex breaks.

Turning to a more directly quantifiable system, we used 405 nm laser microirradiation, without additional photosensitisers, as a sensitive and quantitative method to assess the relative contribution of the ligand-binding PBZ, UBZ and PIP motifs to the localisation and retention of SNM1A to complex DSBs. Direct 405 nm laser irradiation

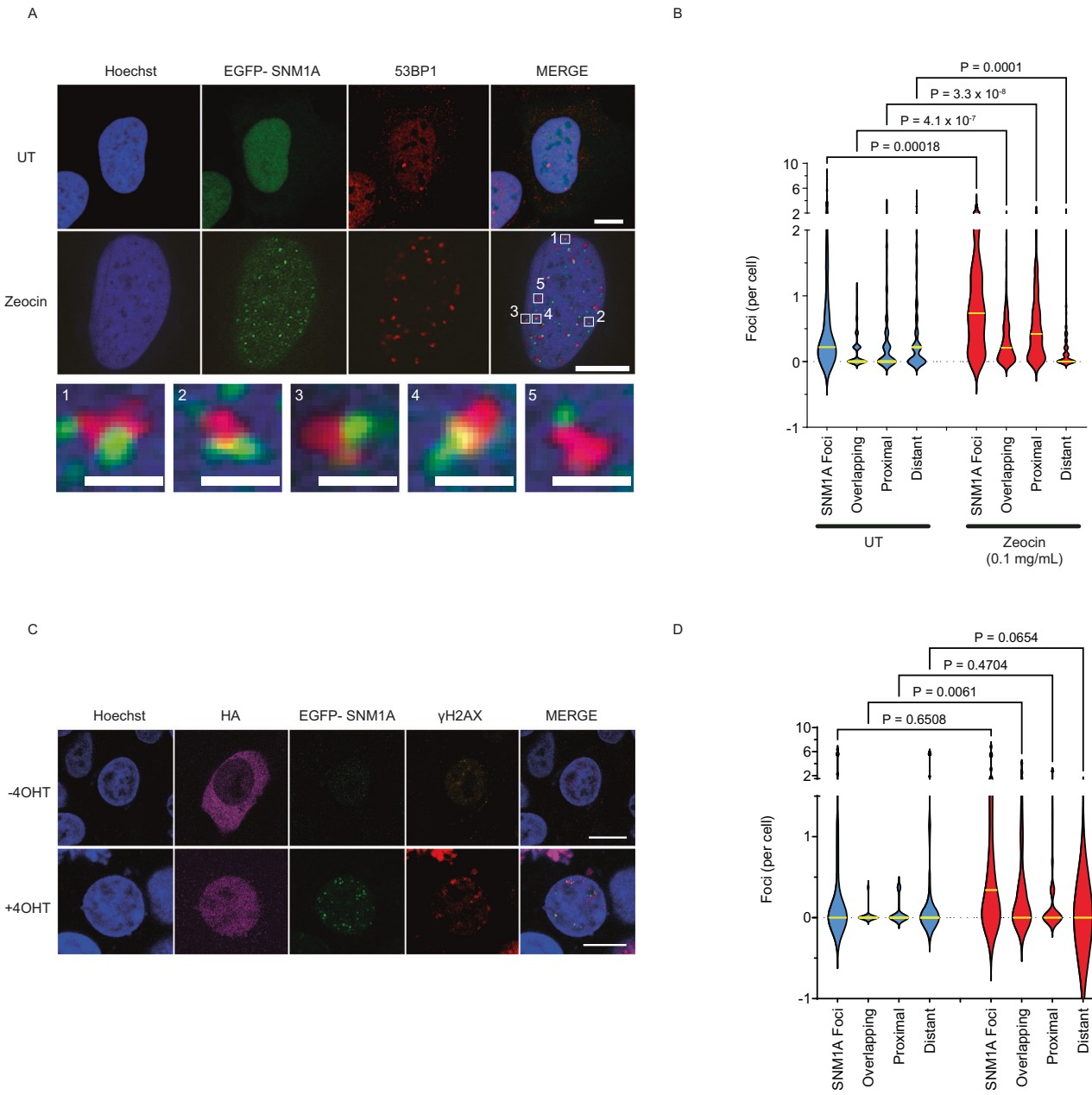

**Fig. 2 | Formation and proximity of SNM1A and 53BP1 foci in response to Zeocin-induced DNA damage. A** EGFP-SNM1A (green) and 53BP1 (red) in untreated (UT) and Zeocin-treated cells (0.1 mg/mL, 2 h) showing the co-proximal accumulation of these two markers at sites of Zeocin-induced DNA damage. Staining with Hoechst 33258 (Hoechst - blue) was used to define nuclei. EGFP-SNM1A expressing cells were fixed and stained with anti-53BP1 (scale bars = 10 μm). Magnified views (bottom row) of select regions of interest of the co-proximal EGFP-SNM1A (green) and 53BP1 foci (red) showing the proximity of these foci (scale bars = 1 μm). **B** Quantification of SNM1A foci and distance to 53BP1 foci showing overlapping (foci are less than 5 pixels, being 0.36 μm, apart), proximal (6–15 pixels, 0.36 to 1.07 μm, apart) and distant (greater than 15 pixels or 1.07 μm apart). Data is from *n* = 3 biological repeats counting 241 and 232 cells for UT and Zeocin respectively. **C** Cells transfected with pBABE HA-*AsiSI*-ER plasmid and treated (500 nM, 4 h) or not treated with 4-hydroxytamoxifen (+ 4OHT and − 4OHT respectively), which induces the HA-tagged AsiSI nuclease to enter the nucleus and induce 'clean' genome-wide sequence-specific DSBs. Cellular location can be seen in purple ("HA") using an anti-HA antibody to the HA-tagged AsiSI nuclease (scale bars = 10 μm). EGFP-SNM1A (green) and DSBs as defined by γH2AX foci (red) are further quantified in (**D**) based on proximity between EGFP-SNM1A and γH2AX as detailed above and described in the methods. Data is from *n* = 3 biological repeats, counting 22 and 30 cells for − 4OHT and + 4OHT respectively. *P* values were calculated with the Kruskal-Wallis test (post-hoc Dunn's multiple comparison test). Source data are provided as a Source Data file.

efficiently induces a high yield of single- and double-strand breaks associated with chemically modified termini, i.e., it acts as an effective surrogate for the damage-induced by ionising radiation and radiomimetic drugs[38]. The EGFP-SNM1A foci in the damage tracks were proximal with 53BP1 foci produced, consistent with the efficient induction of DSBs by microirradiation (Fig. 4A). Examination of the kinetics of recruitment of EGFP-SNM1A in these cells revealed rapid accumulation of EGFP-SNM1A at damage sites, with apparent maximal accumulation within ~10 min of irradiation (Fig. 4B, C). Having established a system to measure the dynamics of EGFP-SNM1A at the sites of

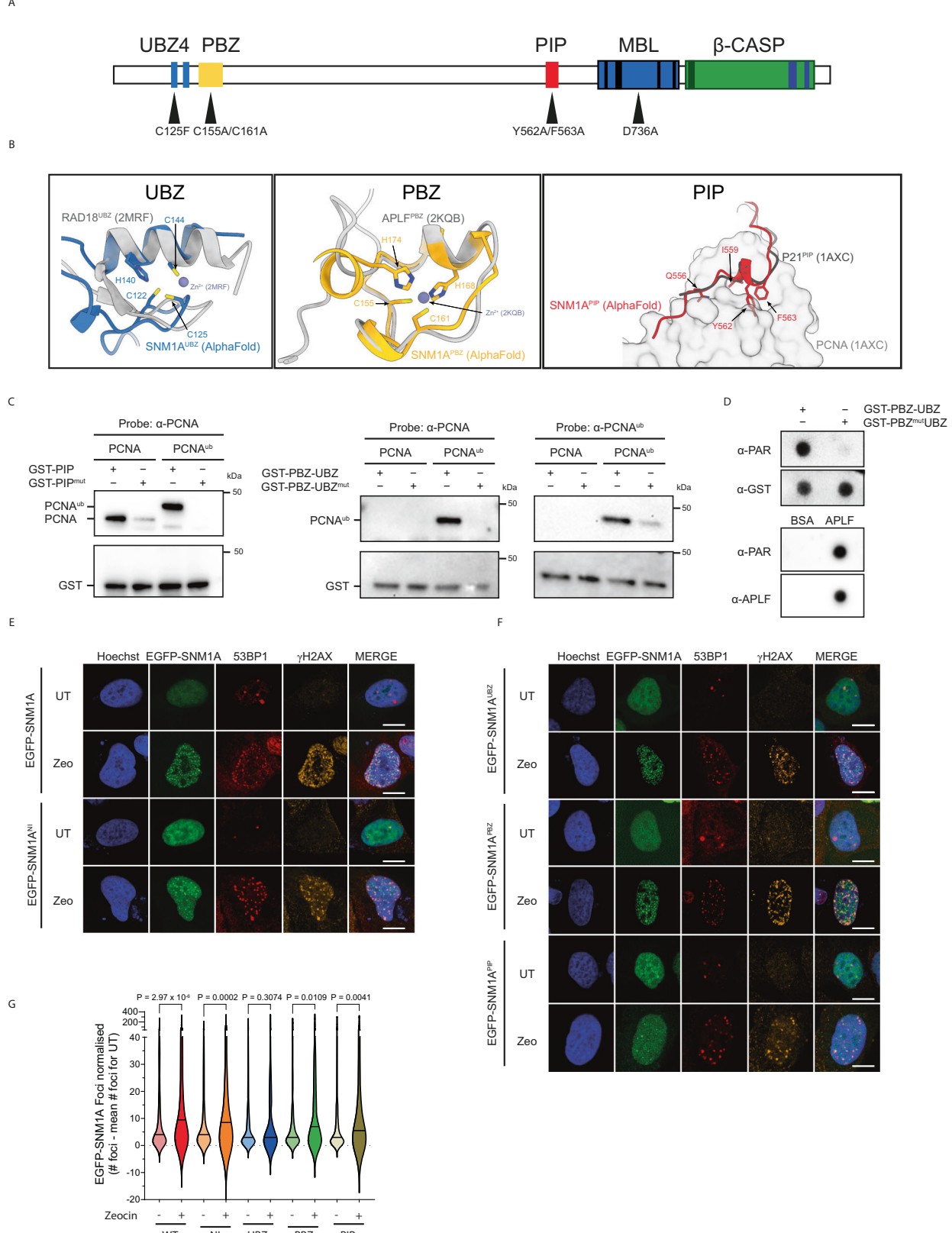

complex DSBs, we examined the impact of inactivation of the catalytic active site of SNM1A on recruitment dynamics. As with its ability to form foci following Zeocin damage, the EGFP-SNM1A[NI] protein (SNM1A[D736A]) was recruited to laser stripes with kinetics indistinguishable from the wild-type protein, confirming that the catalytic activity of SNM1A can be separated from its recruitment to and

retention at complex DNA breaks (Fig. 4B, C). Next, we investigated the possibility of cell cycle phase dependency on the recruitment of EGFP-SNM1A to complex DSBs. EGFP-SNM1A was rapidly recruited to laser-induced damage in G1-, S- and G2-phase cells (Fig. 4D, E), implying a role for SNM1A in processing complex DNA breaks that is sustained throughout the cell cycle.

**Fig. 3 | SNM1A contains conserved ubiquitin-binding, PAR binding and PCNA interacting motifs. A** Domains and motifs of SNM1A, including residues predicted to disrupt ligand binding for each motif in the UBZ4 (ubiquitin-binding zinc-finger 4; C125F), PBZ (PAR-binding zinc-finger; C155A/C161A), PIP box (PCNA-interacting peptide; Y562A/F563A). **B** Structural alignment of the UBZ (blue), PBZ (yellow) and PIP (red) domains using AlphaFold 2.0 overlaid against their respective equivalents (in grey) from RAD18 (PDB 2MRF), APLF (PDB 2KQB) and P21 (PDB 1AXC) respectively. Highlighted amino acids reflect key conserved motifs which coordinate the $Zn^{2+}$ for UBZ and PBZ, and fit into grooves within PCNA (1AXC) for the PIP box. **C** GST-tagged SNM1A peptides were used to pull-down PCNA and lysine-164 monoubiquitinated PCNA (PCNA$^{ub}$). **D** Slot blot analysis of SNM1A-GST peptides indicates wild-type GST peptides spanning the PBZ and UBZ motifs (GST-PBZ-UBZ) are able to bind PAR chains (top panel), whereas mutations in the PBZ (GST-PBZ$^{mut}$-UBZ) do not. GST acts as a loading control, APLF was used as a positive control for PAR-chain binding and BSA as a negative control (bottom panels). **E** Following Zeocin treatment (0.1 mg/mL, 2 h) SNM1A (green), 53BP1 (red) and γH2AX (yellow) foci form and become co-proximal at sites in both wild-type EGFP-SNM1A and also the nuclease inactive (NI) D736A mutant (EGFP-SNM1A$^{NI}$) compared to untreated controls (UT). **F** Protein harbouring substitutions of key residues in the UBZ, PBZ and PIP motifs (substitutions as shown in panel (**A**)) show an altered recruitment of EGFP-SNM1A to Zeocin-induced damage (Zeocin treatment of 0.1 mg/mL, 2 h). **G** Quantification of these damage response EGFP-SNM1A foci (data represents three biological repeats, cells counted from left to right were: 322, 476, 238, 212, 131, 202, 148, 359, 407, 523). Scale bars in (**E**) and (**F**) are 10 μm. Source data are provided as a Source Data file. Blots in C and D are representative of three repeats. Images in (**E**) and (**F**) are representative of data in (**G**).

## The PBZ, UBZ and PIP box domains collectively mediate recruitment and retention of SNM1A at complex breaks

We then analysed the roles of the UBZ, PBZ and PIP box motifs in the recruitment of SNM1A to complex DSBs (Fig. 5A, B). For both EGFP-SNM1A$^{PBZ}$ and EGFP-SNM1A$^{PIP}$, delayed recruitment was observed and these proteins never achieved the same local concentration at the laser site observed for the wildtype SNM1A. By contrast, the EGFP-SNM1A$^{UBZ}$ exhibited a near-normal initial rate of recruitment, though not reaching the levels observed for wildtype SNM1A, followed by gradual loss from the laser sites. EGFP-SNM1A$^{PIP}$ exhibited the most dramatic initial defect in laser stripe recruitment, again suggesting that PCNA interaction is particularly important for SNM1A recruitment to complex breaks (Fig. 5A, B; representative movies are shown in Supplementary Movie 1). Overall, these observations suggest a key role for the PIP box in the initial recruitment of SNM1A to laser damage sites, with the PBZ also contributing to initial recruitment, and that the UBZ motif acts to stabilise the recruited protein at complex DSBs.

We also investigated the kinetics of PCNA recruitment to complex DSBs produced by the 405 nm laser, creating a cell line that stably expressed both EGFP-SNM1A and an anti-PCNA RFP-tagged nanobody[39] (Chromobody®, anti-PCNA VHH fused to red fluorescent protein). We observed that PCNA is recruited to the sites of Zeocin-induced foci and to the sites of laser stripes, where it colocalises with EGFP-SNM1A (Supplementary Fig. 11A, B, C with representative movie shown in Supplementary Movie 2). PCNA recruitment to DSBs precedes that of EGFP-SNM1A by several minutes (Supplementary Fig. 11B, C), occurs in any phase of the cell cycle (Supplementary Fig. 11D) and PCNA recruitment is not delayed or reduced in SNM1A- cells (Supplementary Fig. 11E). These observations imply a key role for PCNA in attracting EGFP-SNM1A to complex DSBs.

To further examine the role of the UBZ, we examined the recruitment kinetics of EGFP-SNM1A in cells depleted for the key E3 ubiquitin ligases involved in DSB repair. These include RAD18 (which is an established E3 ligase for the ubiquitination of PCNA)[40], RNF8 and RNF168 which mono- and poly-ubiquitinate histone H2A, respectively, in response to DSBs. Depletion of these three E3 ligases using siRNA revealed that only depletion of RAD18 impacted SNM1A recruitment to laser damage, and the defect was kinetically similar to mutation of the UBZ domain since it led to wild-type-like initial recruitment kinetics of SNM1A to the stripes, but lower overall levels of accumulation at these sites (Supplementary Fig. 11F). The only known target of RAD18 E3 ligase activity is lysine 164 of PCNA, and indeed a wild-type SNM1A UBZ motif directly mediates interaction with PCNA$^{ub}$ in vitro (Fig. 3C). However, we were unable to detect monoubiquitination of lysine 164 by immunoblotting in whole cell extracts, using a ubiquitin-specific antibody raised against this epitope, likely because the fraction of PCNA which is ubiquitinated in these cells is low. A functional ubiquitin response is clearly required for the efficient recruitment and retention of SNM1A at the sites of laser damage, since pre-treatment of cells with MG-132, an agent that exhausts the cellular free ubiquitin pool by proteasome inhibition[41], dramatically reduced recruitment of EGFP-

SNM1A to laser stripes (Supplementary Fig. 11G). Nonetheless, to definitively address this point, we employed edited 293 T cells where lysine 164 has been substituted with an arginine residue (K164R)[42]. Here, EGFP-SNM1A recruitment to laser stripes was delayed in a manner that phenocopies mutation of the UBZ motif (compare Fig. 5C to Fig. 5A), while PCNA recruitment was unaffected by mutation of this residue, as determined by Chromobody® detection (Fig. 5D).

To examine the interplay and interdependence of the three key conserved motifs involved in SNM1A break localisation, we created double-mutations in the PBZ and UBZ, PBZ and PIP box, and UBZ and PIP box motifs and a form of SNM1A triply mutated in the PBZ, UBZ and PIP box. Analysis of the dynamics of recruitment and retention of the double- and triple-mutated forms of SNM1A reveals that mutation of either the PBZ or UBZ motifs together with the PIP box essentially eliminated SNM1A recruitment to and retention at laser stripes, where dual mutation of the PBZ and UBZ drastically reduced recruitment and retention of EGFP-SNM1A to stripes (Fig. 5E). Accordingly, mutation of all three motifs eliminates recruitment (Fig. 5E).

To explore the role of the PBZ in recruitment of SNM1A to complex DSBs, we exploited, Olaparib which competitively inhibits PARP1 and also produces PAR chain-shielding in cells through trapping the PARP enzyme during catalysis[43]. Pre-treatment of cells with Olaparib led to a reduced rate of recruitment of EGFP-SNM1A to sites of laser damage (Fig. 5F). Combining Olaparib treatment with the EGFP-SNM1A$^{PIP}$ mutant led to an abrogation of recruitment and retention reminiscent of the double substitution mutations in the PBZ and PIP motifs (Fig. 5G). This observation implies that interaction with PAR chains is important for efficient initial recruitment of SNM1A to laser damage. The combined results demonstrate that several conserved motifs work collectively to orchestrate the initial recruitment (PBZ and PIP) of SNM1A to complex breaks and that the UBZ motif is important for SNM1A retention at these breaks, likely in a manner involving association with PCNA$^{ub}$.

Since multiple motifs in SNM1A act in concert to recruit and retain SNM1A at sites of complex DNA damage, we investigated whether interactions with their cognate ligands impact on the exonuclease activity of SNM1A. Although the core catalytic domain of SNM1A is formed by the C-terminally located MBL-β-CASP fold, we wanted to investigate whether the N-terminal motifs that regulate the damage localisation of SNM1A (PBZ, UBZ and PIP) and of SNM1A impact on catalytic activity. To do this, we used a kinetically preferred substrate of SNM1A[44], a single-stranded 21-nucleotide (21-nt) oligonucleotide, bearing a 5'-phosphate group and radiolabelled at its 3'-end. When incubated with full-length SNM1A (which was purified from human cells harbouring the ligand-binding motif substitutions utilised in the preceding cellular studies) all forms of the protein retained nuclease activity, with the caveat that purification of full-length SNM1A yields small quantities of protein and some variation in activity from preparation to preparation was observed (Supplementary Fig. 12A).

We next tested the impact of PCNA and PAR chains on the activity of full-length SNM1A. Titration of PCNA followed by the analysis of the digestion activity over a time course of one hour revealed that PCNA

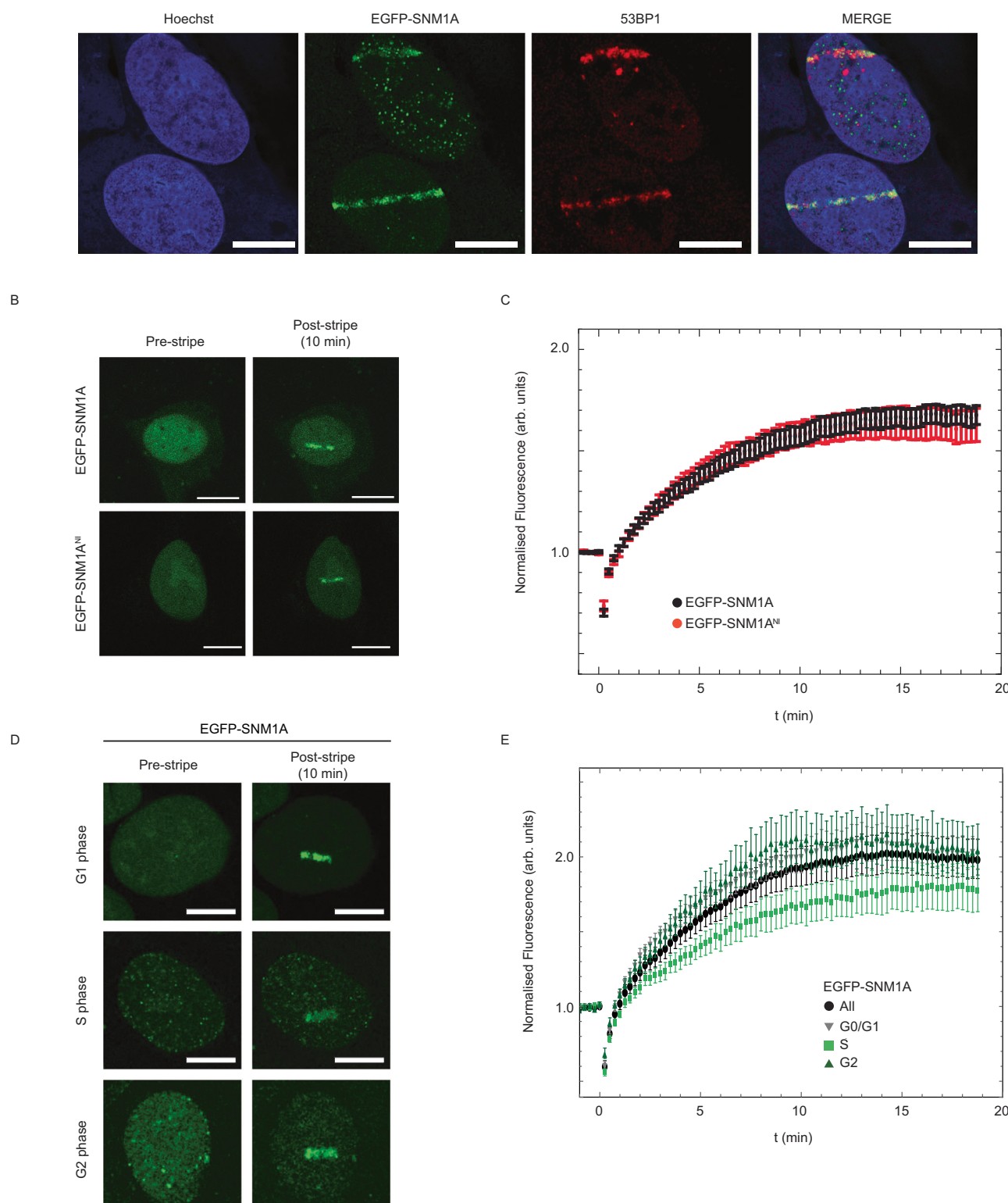

**Fig. 4 | SNM1A is rapidly recruited to laser-induced DNA damage. A** EGFP-SNM1A (green) is recruited to localised laser-induced DNA damage (laser stripes) where it becomes co-proximal with 53BP1 (red). Staining with Hoechst 33258 (blue) was used to define nuclei. **B** To analyse the recruitment of EGFP-SNM1A cells were monitored over 20 min and the fluorescence intensity at the sites of laser striping measured (see "Methods"). Both wild-type EGFP-SNM1A and nuclease inactive EGFP-SNM1A$^{NI}$ were recruited to laser damage with similar kinetics and intensity (quantified in panel (**C**), where $n = 15$ and 11 for WT and NI respectively. Data is the mean and error is SEM). **D** Recruitment of EGFP-SNM1A was similar in G0/G1, S and G2-phases of the cell cycle (quantified in panel (**E**). where $n = 40$, 28 and 14 for G0/G1, S and G2 respectively. Data is the average and the error is the SEM). Scale bars in (**A**), (**B**), and (**D**) are 10 μm. All post-stripe cell images represent the 10 min time point. Source data are provided as a Source Data file.

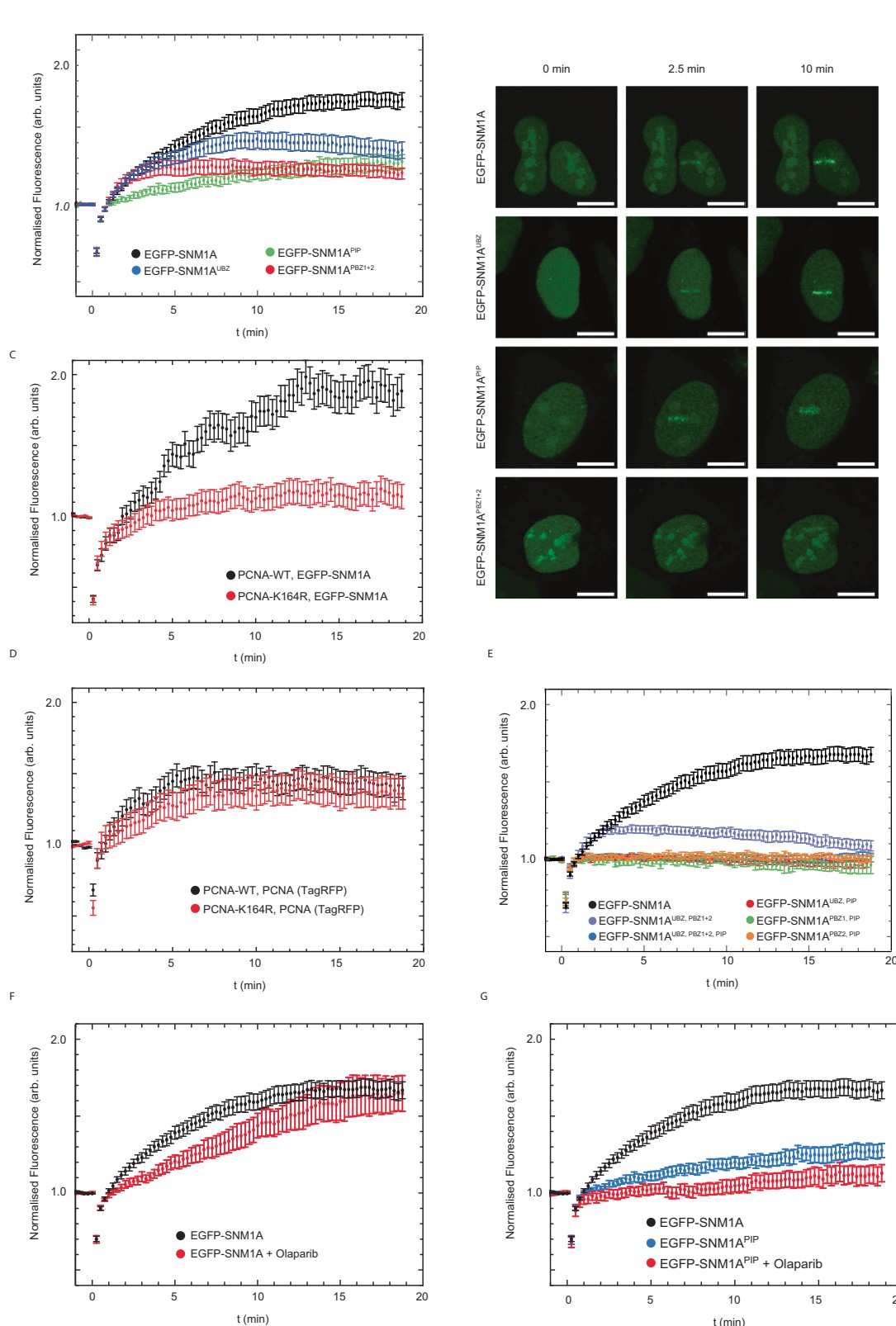

did not enhance the activity of SNM1A, with some decrease in digestion being observed (Supplementary Fig. 12B). To determine whether this was a direct effect mediated by interaction of PCNA and SNM1A, we repeated this experiment with N-terminally truncated SNM1A (ΔN-SNM1A, residues 697–1040) retaining the MBL-β-CASP fold (and exonuclease activity), but lacking the PIP box. ΔN-SNM1A activity was also

reduced in the presence of PCNA (Supplementary Fig. 12B) suggesting that perturbation of nuclease activity in the presence of PCNA is possibly a result of reduced SNM1A access to the 5'-terminus of its DNA substrate in the presence of PCNA. Next, we incubated SNM1A with PAR chains, or performed reactions in a system that produces PAR chains in situ by pre-incubating poly-ADP-ribose polymerase with

**Fig. 5 | The UBZ, PBZ and PIP box motifs of SNM1A play a role in protein recruitment to sites of laser-induced DNA damage. A** The recruitment in U2OS cells of EGFP-SNM1A containing substitutions in the UBZ (C125F), PBZ (PBZ1 = C155A and PBZ2 = C161A) and PIP box (Y562A and F563A) that ablate direct ligand-binding (see Fig. 3) were monitored following laser-induced damage, representative images shown in panel (**B**) where scale bars are 10 μm. For (**A**) n = 15, 16, 14 and 10 for WT, UBZ, PBZ[1+2] and PIP respectively. Data is the average and the error is the SEM. **C** Transiently expressed EGFP-SNM1A recruitment to laser stripes in HEK293T and 293T-K164R cells (cells expressing a mutation in the ubiquitin-modified PCNA residue which are unable to be ubiquitinated on K164). Data is the mean, n = 20 for both conditions, the error is SEM. **D** Recruitment of PCNA to laser stripes as measured through a transiently expressed RFP-PCNA Chromobody® (Tag-RFP) in HEK293T and 293T-K164R cells. The data is the average of n = 20 for both conditions where the error is SEM. **E** Double or triple co-substitution of key residues in the UBZ, PBZ and PIP box were used to investigate the cooperative contribution of these SNM1A motifs to laser stripe recruitment in U2OS cells. Data is an average (error = SEM) where n = 15 for WT, n = 15 for UBZ + PBZ[1+2], n = 10 for UBZ + PBZ[1+2] + PIP, n = 10 for UBZ + PIP, n = 6 for PIP + PBZ[1], n = 18 for PIP + PBZ[2]. **F** Olaparib treatment (5 μM, 20 h) in U2OS cells reduces EGFP-SNM1A recruitment to laser stripes (n = 15 for both WT and WT + Olaparib, data is displayed as the average and the error is SEM). **G** The effect of combining PIP box mutation with Olaparib (5 μM, 20 h) treatment on laser stripe recruitment shown in U2OS cells. Data is the average of 15, 10 or 8 cells for WT, PIP and PIP + Olaparib respectively. Error is SEM. Source data are provided as a Source Data file.

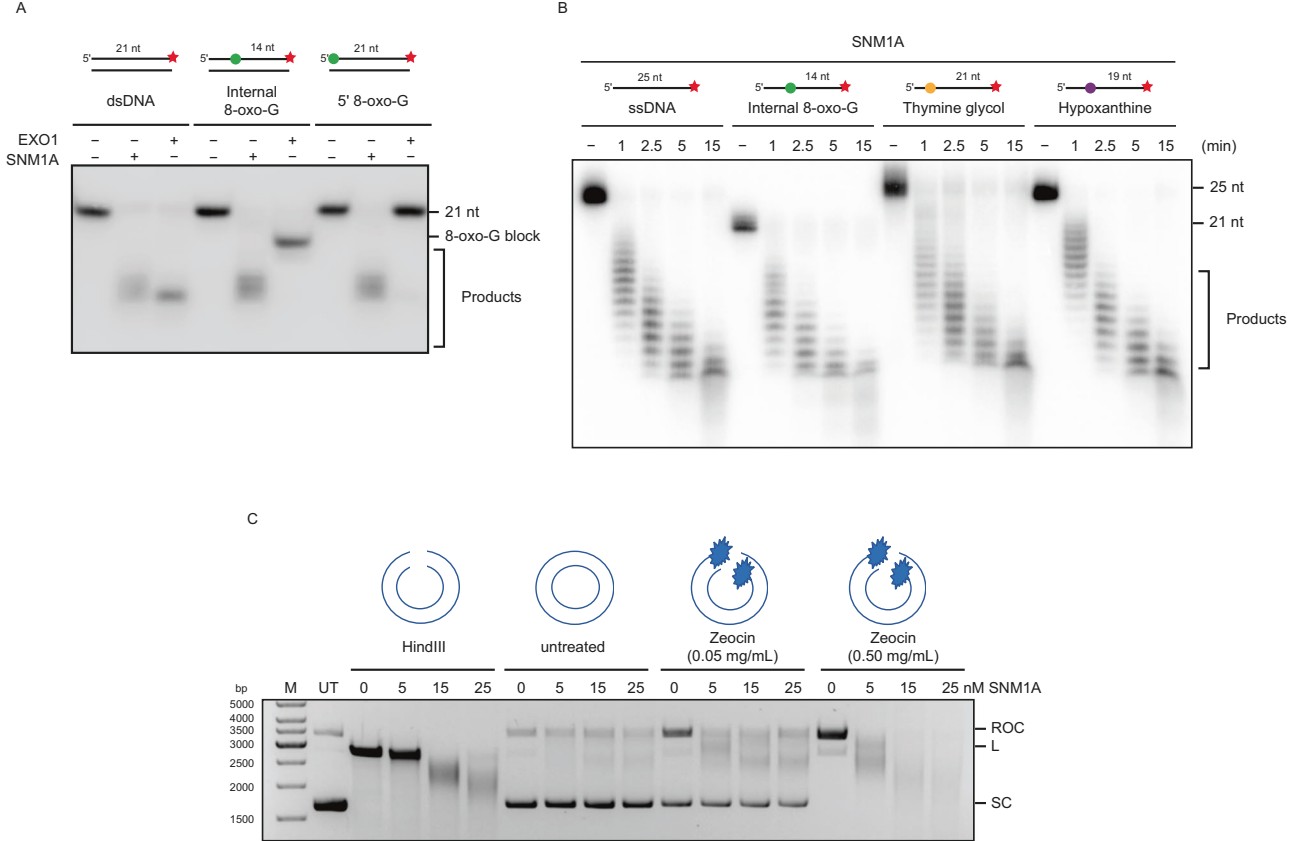

**Fig. 6 | Purified SNM1A can digest complex DNA lesions in vitro. A** Purified SNM1A and EXO1, can digest 3′ radiolabelled (red star) dsDNA in vitro. SNM1A and EXO1 were also incubated with oligonucleotides with either an internal or a 5′-located 8-oxoguanine (8-oxo-G, green circle). SNM1A can digest past the altered 8-oxo-G bases but EXO1 does not. Representative gel of n = 2 repeats. **B** SNM1A can digest substrates containing several major oxidative DNA damage products, including 8-oxo-G (green circle), Thymine glycol (yellow circle) and Hypoxanthine (purple circle). Representative image of six repeats. **C** Untreated (UT) or HindIII linearised pG46 plasmid was treated in the presence or absence of Zeocin (0.05 mg/mL and 0.5 mg/mL, 1 h) in a pharmacologically-relevant iron-catalysed reaction to induced complex DNA breaks. Plasmids were treated with purified SNM1A (0 to 25 nM, 1 h) and the resulting DNA products resolved on an agarose gel, stained with Ethidium bromide and visualised. SC = supercoiled DNA; ROC = Relaxed Open-Circular DNA; L = Linear DNA. The image is representative of three repeats.

NAD+ and the DNA substrate with the subsequent addition of purified full-length SNM1A to the reaction. The activity of SNM1A was not modulated by the presence of PAR chains (Supplementary Fig. 12C). Together, these biochemical analyses suggest that the interactions mediated by the PBZ, UBZ and PIP box motifs are principally important for recruiting and retaining SNM1A to DNA damage, rather than modulating the activity of the enzyme.

### SNM1A can process DNA containing oxidised lesions

It was of interest to investigate whether the requirement of SNM1A for efficient repair of complex DSBs relates to the ability of SNM1A to process DNA containing lesions induced by radiation and radiomimetic damage, analogous to its established role in resecting DNA containing ICLs. We initially employed 21-mer double-stranded oligonucleotides that contain an 8-oxoguanine (8-oxo-G) residue on the substrate strand, one of the major oxidative lesions induced by radiation at complex breaks[45], either located centrally or at the 5′-terminus. Incubation of these substrates with ΔN-SNM1A revealed that its exonuclease activity can traverse the lesion without pause or arrest (Fig. 6A). This striking activity contrasts starkly with that of human Exonuclease 1 (EXO1), one of the major exonucleases involved in performing DNA extensive end resection in preparation for homologous recombination, which is quantitatively arrested by an 8-oxo-G lesion, regardless of whether it is located terminally or centrally in the

substrate (Fig. 6A; baseline activities of SNM1A and EXO1 were normalised on an undamaged substrate, Fig. 6A, first 3 lanes). We examined the capacity of SNM1A to process additional oxidative lesions commonly associated with complex DSBs. Examining related substrates that contain thymine glycol and hypoxanthine bases reveals that SNM1A efficiently digest ssDNA substrates containing these lesions, with kinetics comparable to those observed on a native, undamaged template (Fig. 6B), and again, we observed thymine glycol acted as a complete block to EXO1 digestion (Supplementary Fig. 12D). Finally, we determined whether SNM1A can process DNA substrates containing Zeocin-induced DNA breaks in vitro. We employed two doses of Zeocin in an iron-metal catalysed reaction that mimics the activation of Zeocin in cellular conditions[46], we introduced SSBs (0.05 mg/mL Zeocin; producing relaxed open-circular DNA) and a mixture of SSBs and DSBs (0.5 mg/mL Zeocin; producing relaxed open-circular DNA and linear DNA forms) into plasmid DNA. SNM1A was capable of extensively digesting the nicked and linear forms of DNA molecule in a concentration and time-dependent manner (Fig. 6C and Supplementary Fig. 12E, F), consistent with a capacity to exonucleolytically process the chemical modifications induced by radiomimetics at the sites of the DNA breaks they induce.

## Discussion

Disruption of SNM1A in human cells reveals an unanticipated phenotype, a marked sensitivity to radiomimetic drugs and IR treatment, that is associated with delayed resolution of DSBs. SNM1A is recruited to the complex DNA breaks induced by radiomimetics, IR and long-wave laser damage within minutes of damage induction. The EGFP-SNM1A recruited to radiomimetic and laser stripe damage lie either adjacent to or in some cases overlap with, the damaged chromatin nanodomain markers such as 53BP1 and γH2AX. The precise nature of these distinct classes of foci is currently unclear and it will be of interest to characterise these in future studies. Multiple potential ligand-interacting motifs are present in the N-terminal region of SNM1A, proximal to the MBL-β-CASP domain which forms the catalytic core and active site[15]. A systematic examination of the role of the SNM1A putative interaction and recruitment motifs revealed an important role for the PIP box and PBZ motif inefficient initial recruitment of SNM1A to sites of complex breaks. This finding implies that the interaction of SNM1A with PCNA (which is itself very rapidly recruited to laser-induced damage, as demonstrated here and by others[47]) and PAR chains is critical for the timely recruitment of SNM1A to laser damage. This conclusion is consistent with the reported ICL-induced interaction of SNM1A with PCNA[18]; the data presented here demonstrates that SNM1A interacts with PAR chains in vitro and a PBZ-dependent manner. Moreover, trapping and inhibiting poly-ADP-ribose polymerase (PARP) with the PARP inhibitor Olaparib phenocopies mutation of the PBZ motif, i.e., a reduction in initial SNM1A recruitment to laser damage is observed. Indeed, co-disruption of the SNM1A PIP box with the PBZ completely abrogates recruitment of SNM1A to the sites of laser and Zeocin-induced breaks and, consistently, so does mutation of the PIP box when combined with Olaparib treatment.

SNM1A contains a UBZ motif, adjacent to its PBZ motif. Like its PIP box motif, the UBZ motif is reported to be important for targeting SNM1A to the ICL damage foci during S-phase[18]. In response to laser-induced damage, mutation of the UBZ motif produced a phenotype distinct from that observed by mutation of the PBZ or PIP box. Importantly, this result suggests that mutations in the neighbouring UBZ or PBZ motifs do not mutually impact the function of the other motif. UBZ mutant SNM1A is recruited with near-normal initial kinetics to laser damage but fails to accumulate to the same final level. A screen of ubiquitin E3 ligases that are known to deposit ubiquitin at sites of DNA breaks, and which are therefore candidates for providing the ligand for the interaction with the SNM1A UBZ motif, reveals that RAD18 loss phenocopies mutation of the UBZ, leading to reduced

overall levels of accumulation of SNM1A at laser stripes. PCNA (lysine 164) remains the only known target for RAD18 ubiquitination[48], though we were unable to detect PCNA ubiquitination following IR or Zeocin treatment, either in whole cell extracts or by performing immuno-precipitations with an antibody directed against monoubiquitinated PCNA. This is likely because the fraction of PCNA that is ubiquitinated is low. A role for RAD18 in DSB and replication fork repair that relies on interaction with the SLF1 and SLF2, and is important for recruitment of the SMC5/6 complex to chromatin, has been previously reported[49]. However, this RAD18-subcomplex is recruited to damage by RNF168, which is dispensable for SNM1A recruitment to DSBs (Supplementary Fig. 11F), suggesting that the RAD18-SLF1-SLF2 complex does not play a major role in recruiting SNM1A to DSBs. Moreover, as cellular PCNA K164 is required for normal SNM1A retention at laser stripes (Fig. 5C), and the UBZ motif of SNM1A directly interacts with ubiquitinated PCNA[ub] (Fig. 3C), the evidence that ubiquitinated PCNA acts to retain SNM1A at the sites of such damage is robust. The phenotypes of cells disrupted for homologues of SNM1A have been examined in multiple organisms, ranging from yeasts to humans, and has revealed a conserved role in the repair of ICLs. However, where the sensitivity and a response to IR and radiomimetics has been examined in vertebrate cells, only human cells have been implicated in response to these forms of damage. In the case of mouse ES cells and chicken DT-40 cells, no marked sensitisation to IR was observed[50–52]. However, work from Richie and colleagues has shown that human SNM1A forms subnuclear foci following IR treatment[53], and that these foci localise with 53BP1 and Mre11, providing early evidence for a role for SNM1A in the repair of radiation-induced DSBs. A potential explanation for these inter-species differences rests with the fact that vertebrates harbour (at least) three SNM1 paralogues, SNM1A, SNM1B/Apollo and SNM1C/Artemis. Based upon substantial similarities in their 5′-3′ exonuclease catalytic activities, SNM1B might plausibly play a related or redundant role in the processing of radiation-induced DSBs. Indeed, multiple reports indicate that loss of murine SNM1B is associated with increased radiosensitivity[20,54]. Therefore, in mice the roles of SNM1A and SNM1B in the repair of complex DNA breaks might only be revealed once such redundancy or species-specific prioritisation of their roles has been systematically examined. Moreover, Artemis/SNM1C deficient cells are IR sensitive and Artemis/SNM1C plays an established role in the removal of several end-blocking chemical modifications during repair of complex breaks. Strikingly, disruption of the SNM1A homologue in budding yeast, Pso2, together with inactivation of the nuclease Mre11 (via an *mre11-H125N* mutation) leads to a marked sensitivity to IR, suggesting that these two nucleases may play, at least partially, redundant roles in processing complex DNA ends[55]. Notably, like the *pso2* and *mre11* single mutants, *pso2 mre11-H125N* double mutant cells did not display any strong defects in the repair of 'clean DSBs' induced by the HO-endonuclease, analogous to the situation we report here for SNM1A in the I-SceI reporter assay (Supplementary Fig. 8), despite the fact we are able to observe a degree of recruitment to clean DSBs induced by an endonuclease (Fig. 2B). This suggests that a key function of Pso2 and SNM1A (and a known major function of MRE11) is in processing of chemical modifications to DNA at 'dirty' break termini before their repair, rather than direct participation in the DSB process per se. MRE11 is also implicated in the initiation of DSB resection and repair in cells by clearing covalently linked proteins at break termini, for example, the DNA-topoisomerase 1 (Top1) crosslinks induced through camptothecin (CPT) treatment[56,57]. However, we see no evidence of a role for SNM1A in this process, based on the wild type like sensitivity of SNM1A cells to CPT. It is also interesting to note that co-inactivation of Pso2 with Exo1, the only other known major 5′-3′ repair exonuclease in yeast, does not strongly impact on IR sensitivity[55]. This observation implies that Exo1 is not a major factor acting to process chemically modified termini in the absence of Pso2, and vice versa. Our finding that the exonuclease activity of human EXO1, a close functional

relative of yeast Exo1, is highly sensitive to the presence of chemical lesions is consistent with this proposal. It seems likely that both MRE11 (through its 3′-5′ exonuclease or endonuclease activities) and SNM1A (through its 5′-3′ exonuclease activity) are well equipped to deal with a set of the chemical lesions induced at break termini by radiation and play a major role in this important defence against complex DSBs. Therefore, a comprehensive examination of the relationship between all three SNM1 paralogues in mammalian cells, in addition to their contribution relative to other established and putative end-processing nucleases (in particular MRN), is warranted in future studies.

Finally, the identification of a role for SNM1A in processing radiation-induced oxidative modified termini in DSBs raises the possibility that SNM1A inhibition might be used to reduce the doses of radiation treatment required to treat cancer. The results presented here show that the interactions mediated by the PBZ, UBZ and PIP box motifs are principally and collectively important for recruiting and retaining SNM1A to DNA damage, rather than modulating the activity of SNM1A. Thus, targeting the β-CASP-MBL fold domain of SNM1A is likely a preferred mode of inhibition and is the subject of ongoing efforts.

# Methods

## Cell lines
U2OS (American Type Culture Collection: HTB-96), 293FT (Thermo-Fisher Scientific: R70007), HEK293 XRCC4⁻ (a gift from Andrew Blackford) and 293 T PCNA$^{K164R}$ cells (293T-K164R) (a gift of George-Lucian Moldovan) were cultured in D-MEM medium supplemented with 10% foetal bovine serum.

## Creation of SNM1A⁻ cell lines using genome engineering
Zinc finger nucleases (ZFN) and CRISPR-Cas9 genome editing technologies were employed to make stable deletion-disruptions in *DCLRE1A*, the gene encoding SNM1A. CompoZr ZFNs were designed by and purchased through Sigma-Aldrich with a detection and cut site in the *DCLRE1A* gene as follows (cut site in lowercase):

TGCCAGATGCCTTTTtcctcATTGATAGGGCAGAC

Cells were transfected with plasmids (~1.3 μg) containing forward and reverse ZFNs in 10 μL of Lipofectamine 2000 in a total volume of 200 μL OptiMEM media (Sigma). The transfection mixture was incubated (10 min) to allow the complexes to form before being added to $5 \times 10^5$ cells that were freshly plated in 2 mL of medium in wells of a 6-well plate. The medium changed after 12–18 h; after a further 72 h, the cells were analysed for genomic insertions or deletions using the SURVEYOR Mutation Detection Assay (Integrated DNA Technologies). Once a pooled cell population was identified to contain cells with altered genomes, dilution cloning was performed. Once grown, the clones were assayed for alterations to their genomes through genomic PCR. Identified SNM1A⁻ clones were validated by analysing SNM1A protein levels (through SDS-PAGE/Western blot techniques).

CRISPR-Cas9 genome editing was performed using sgRNAs designed using an in-house design tool. The targets for the CRISPR-Cas9 sgRNAs of the *DCLRE1A* gene, which remove the majority of exon 1, are shown in Supplementary Table 1.

The sgRNAs were cloned into CRISPR-Cas9 vectors pX330 and pX458 carrying GFP and RFP fluorescence expression markers respectively.

Once treated with both plasmids (as above for ZFNs), the cells were sorted by FACS to collect individual GFP and RFP positive cells in a well of a 96-well plate. The clones were grown before being analysed for genomic alterations (as above). The SNM1A⁻ cells used are detailed in Supplementary Fig. 1.

## Cell transfections for expression
For laser microirradiation, or analysis of drug-induced DNA damage (foci), cells were transiently transfected with pEGFP-C1 plasmids containing wildtype SNM1A and substitutions in the domains of interest.

Cells ($5 \times 10^5$) were seeded in 35 mm glass bottom μ-Dish dishes (Ibidi). Transfection was performed with ~2.6 μg plasmid in 2 mL media with 10 μL Lipofectamine 2000 (ThermoFisher Scientific). Complexes were allowed to form for 10 min and then added dropwise to the cells.

For the creation of cells stably expressing plasmids containing EGFP-SNM1A or a Chromobody® (Tag-RFP) of RFP-PCNA (Chromotek), ~$5 \times 10^5$ cells were seeded in wells of a 6-well plate and immediately transfected with the desired plasmid (2.6 μg) with Lipofectamine 2000 (2 μL) in a total volume of 200 μL. A reduced volume of Lipofectamine 2000 was used to reduce the toxicity during transfection. After 3 h, the media was changed and the cells were grown for 24 h. The cells were sorted and single cells expressing the fluorescence marker associated with the plasmid were plated out in a 96-well plate and allowed to grow until 50–75% confluent. The cells were transferred to 25 cm² flasks and allowed to grow further. Once 60–80% confluent, cells were analysed for fluorescence and clones that exhibited an adequate level of fluorescence were further analysed by microscopy, SDS-PAGE, and western blot protocols to establish the desired expression and expected phenotype of the clones. For the nuclease domain, PIP box, PBZ and UBZ substitution mutants, standard site-directed mutagenesis techniques were used to introduce the indicated sequence changes (see Fig. 3).

Cells that had successfully integrated the pEGFP-C1-SNM1A plasmid into their genome were first selected with kanamycin before being sorted by FACS into single wells in a 96-well plate, and grown up as a clonal population.

For analysis of clean DNA-DSB the pBABE HA-*Asi*SI-ER plasmid (a kind gift from Monika Gullerova, Sir William Dunn School of Pathology, Oxford) was transiently transfected into a clone of 293FT SNM1A⁻ cells that stably express EGFP-SNM1A using the protocol listed above. Nuclear localisation of the AsiSi endonuclease was achieved by treatment with 500 nM 4-hydroxytamoxifen (4OHT) for 4 h. Location of the HA-tagged *Asi*SI was performed by an antibody directed against HA (see Supplementary Table 2)[30].

## siRNA Transfections
Cells were treated with 20 nM siRNAs in HiPerFect transfection reagent (Qiagen) using a fast-forward transfection protocol then further siRNA was added 24 h later, as per the manufacturer's protocol. Sequences of the siRNAs used in this study can be found in Supplementary Table 3. BRCA2, RNF8 and RNF168 siRNAs were from Life Technologies.

RAD18 siRNA was a smart pool from Horizon (catalogue number L004591-00-0005).

## Cell treatments with DNA damaging drugs and radiation
Cell treatments, unless otherwise stated in figure legends, were; Cisplatin (CDDP, Teva pharmaceutical industries Ltd., Eastbourne, UK. Cat # 51642169) is a concentrate for clinical infusion, including 1 mg/mL cisplatin with sodium chloride, hydrochloric acid/sodium hydroxide (for pH adjustment). If required, CDDP was diluted in PBS, treatments were for 4 h. Zeocin (Invitrogen, ThermoFisher Scientific, Cat # R25001) was diluted in PBS. Treatments were at 0.1 mg/mL for 2 h, unless otherwise listed in figure legends. Ionising radiation-induced damage (IR) was induced using a caesium 137 source. SJG-136 (a gift from John Hartley, UCL) was dissolved in H₂O (1 μM) and treatments were performed continuously. Olaparib was made up in PBS (10 mM) and clonogenic treatments were continuous. For striping data, Olaparib treatment was 5 μM for 20 h. Methanol-free aqueous formaldehyde (16% (v/v) (Taab Laboratories Equipment) was always freshly diluted in PBS and used immediately, treatments were for 2 h. UV-C treatments were performed using a Stratagene, UV stratalinker 2400 machine with UV-C lamps. Before UV treatment, the media was removed from the dishes, the cells were dosed with UV-C and fresh media was added. Hydrogen peroxide was diluted in PBS from a stock 30% solution, treatments were continuous (BDH, Cat#BDH7741-1). MG-132 (Med Chem Express) was dissolved in DMSO (1 mM) and cells were

pre-treated with 5 µM for 90 min. Camptothecin (Cambridge Bioscience) was dissolved in DMSO treatments were for 1 h.

## Colony counting/clonogenic assays

Clonogenic assays were performed in 10 cm tissue culture dishes (for U2OS and clones) or T75 cm² tissue culture flasks (293FT cells, due to their reduced colony-forming ability). Cells (1000 per dish for U2OS cells and 2000 per flask for 293FT cells) were seeded in complete media (10 mL and 20 mL respectively) and allowed to attach overnight before being treated with the desired agents. Following treatments (short or continuous) cells were allowed to grow and form colonies for 10 days. Colonies formed were stained with Coomassie R250 (Sigma) and counted on a COLCount Colony Counter (Oxford Optronix). All experiments represent the mean ($\pm$ SEM) of at least three biological repeats of duplicate dishes/flasks for each treatment.

## Structural prediction

ColabFold Google Colabs notebooks were used to predict structures of the UBZ, PBZ and PIP Box domains (https://colab.research.google.com/github/sokrypton/ColabFold/blob/main/AlphaFold2.ipynb)[58]. With default parameters and inputting the sequences: (numbering as for human SNM1A; NCBI /NP_001258745.1/): UBZ = 112–148; PBZ = 153–182; and PIP = 552–573. Coordinates of the highest-ranking AlphaFold models were then visualised, aligned to prior structures, and rendered using ChimeraX[59]. Alignments were made to prior structures as follows: RAD18 UBZ – PDB 2MRF[35]; APLF PBZ – PDB 2KQB[36]; PCNA with P21 PIP – PDB 1AXC[37].

## DNA content and cell cycle FACS analyses

An estimation of the distribution of the cell cycle phase of treated cells was performed using BrdU incorporation and DNA content analysis[17]. Following 30 min pre-treatment with 10 µM BrdU cells were harvested and fixed with 70% ethanol on ice for 30 min. Following acid denaturation (2 M HCl) and neutralisation (Na$_2$B$_4$O$_7$) cells were incubated with anti-BrdU and an anti-rat Alexa Flour 488. Following washing, cells were resuspended and nuclei were stained in propidium iodide solution (25 µg/mL). Data was acquired on an Invitrogen Attune NxT flow cytometer and quantification of the phase of the cell cycle was performed post-acquisition using FCS Express software (De Novo Software), example gates used are shown in Supplementary Fig. 5.

## Laser damage striping

DNA damage microscopy experiments were conducted using a Zeiss 780 or Zeiss 880 inverted confocal microscopes using a Plan-APO 63 × 1.40NA oil immersion objective, with the optical zoom adjusted to a projected pixel size of 100 nm, at physiological conditions (37 °C and 5% CO2). Before all experiments, care was taken to adjust the collimation of the 405 nm laser such that the z-focus aligned with the visible lasers (488, 561, and 633 nm) as determined by imaging of 200 nm diameter TetraSpeck beads. The laser power of the 405 nm lasers on both the Zeiss 780 and the Zeiss 880, as measured at the focal plane of a 10 × 0.45NA air objective, was about 5 mW.

DNA damage was induced by defining a rectangular ROI damage site within the cell nuclei with varied lengths (6.5 µm for single-cell experiments and around 200 µm for multi-cell experiments) but a fixed width of 1 µm. The laser damage was then inflicted using the 405 nm at full laser power, a scan speed pixel dwell time setting of 8 µs, and with 25-line iterations. To observe the recruitment of EGFP tagged wildtype and mutant SNM1A to the DNA damage sites, we performed time-lapse experiments for a total of 80 image frames at a frame interval of 15 sec where the 405 nm laser-induced DNA damage was initiated after image frame 5. EGFP emission in these experiments were collected on a GaAsP detector, with a pinhole setting of 1 AU, a bandpass emission setting of 499–544 nm, a pixel dwell time of 1.58 µs, and a line averaging setting of 4.

Quantitative analysis of the active recruitment to the DNA damage site was performed using ImageJ and Mathematica. We extracted the mean fluorescence intensity F(t)Bleach from a region-of-interest (ROI), superimposed on the DNA damage site, relative to the mean fluorescence intensity at a site on the same cell but away from the damage site F(t)Control) as a function of time, from several cells. The data were normalised by the fluorescence intensity F(-) before the induction of the DNA damage such that active recruitment results in a ratio of [F(t)/F(-)]Bleach / [F(t)/F(-)]Control > 1 versus no recruitment or passive recruitment results in a ratio of [F(t)/F(-)]Bleach / [F(t)/F(-)]Control < = 1. The presented normalised data represents the mean ($\pm$ SEM) for at least $n$ = 10 cells from at least three biological repeats.

Cells were scanned in a variety of phases of the cell cycle. We determined the cell cycle phase using multiple approaches. PCNA forms foci during DNA replication, which indicates the cells are in S-Phase. As cells exit S-phase PCNA foci are lost, thus we were able to determine cells in S-phase using PCNA foci presence. To determine the G1- versus G2/M-phase of the cell cycle, we used the brightfield morphology of the cells we were scanning. Cells in G2/M-phase round-up of the microscope slide[60] and can be determined in a higher focal plane than surrounding G1- or S-phase cells.

Antibodies used in this study can be found in Supplementary Table 2.

## Immunoblot (western blot) analysis

Immunoblot (western blot) analyses were performed as described[17]. All secondary antibodies were raised in goats (Dako, Agilent Technologies) and used at 1:5000 in TBS-T (Tris-buffered saline with 0.1% Tween).

## Microscopic analysis of damage-induced subnuclear foci

To assess the number of nuclear foci following treatments, cells were plated in glass bottom dishes (as above) and allowed to attach. Following treatment, cells were fixed for 20 min with 4% Formaldehyde in PBS, blocked with immunofluorescence (IF) blocking buffer (5% horse serum, 1% saponin in PBS) for 1 h before being incubation with primary antibodies at the desired concentrations in IF blocking buffer (overnight, 4 °C). After washing cells three times with PBS, cells were incubated with secondary antibodies (1:500) for 2–4 h in IF blocking buffer, washed a further 3 times with PBS then stained with Hoechst 333258 (1 µg/mL in PBS) for 30 min before a further three washes (all washes and incubations were performed at room temperature unless otherwise stated).

Confocal images were obtained with a Plan APO 63 × 1.40NA oil immersion objective, a pinhole setting of 1 AU, bandpass emission settings of 410–468 nm for Hoechst, 490–544 nm for EGFP, 579–624 nm for RFP or Alexa 568, and 633–695 nm for Alexa 647, a projected pixel dimension of around 110 nm x 110 nm, a pixel dwell time of 1.35 µs, and with a line averaging setting of 2. To ensure sufficient cell numbers (N > 300), images were acquired in a tiled format, either 5 × 5 or 10 × 10 images, corresponding to an image area of about 0.65 mm × 0.65 mm or 1.3 × 1.3 mm respectively. Images were imported into ImageJ and foci were counted using a macro script adjusting for staining levels between experiments. Background fluorescence was determined and controlled by adjusting the laser power to ensure the maximum fluorescence in each channel was below saturation. Processing in image J using the macros script then normalised the fluorescence output using the Huang Dark auto threshold setting as has been previously evaluated[61]. A distribution of whole-cell fluorescence of EGFP- SNM1A is still observed. This method enables us to efficiently detect foci over a broad range of expression patterns, from low-expressing to high-expressing cells. It is noted that due to this approach giving a range of expressions, we followed up with the striping experiments to enable a more controlled study including temporal aspects or recruitment.

The processing and analysing of foci images using imageJ/Fiji macro scripts[62] were preformed as follows. Multi-channel images were split into specific fluorescent channels and then the Hoechst channel was segmented to extract nuclear regions with additional splitting of cellular instances using a watershed filter. The green (EGFP), red (Alexa Fluor 568) and far red (Alexa Fluor 647) channels were filtered with a mild (0.8 sigma) Gaussian blur kernel and then regions corresponding to cells identified through correspondence with the segmentation. From these processed channels, foci were detected in each cell using the Fiji Find Maxima algorithm within the two channels, to create two sets of points for each cell. Each point was analysed for size and also its distance to its nearest neighbour in the comparative channel. Foci were filtered based on their size and distance and then counted and also the intensity was measured in one or more channels for each focus, depending on the experiment, and statistics for each cell established.

To analyse the distance of the DNA damage marker foci (53BP1 or γH2AX) to EGFP-SNM1A foci the image J macro script was used. Cells that were counted in the data for Fig. 2B, D, as well as Supplementary Fig. 10, were identified as having EGFP fluorescence above the background as set by the auto threshold parameter. The foci number in these cells was then used to obtain the average number of EGFP-SNM1A foci. In only EGFP-SNM1A expressing cells (those with and without foci), the number of 53BP1 or γH2AX foci were then counted. In each case, the distance from an EGFP-SNM1A foci was calculated for each other foci and divided into three groups overlapping, proximal and distant. Overlapping foci were those foci which were within 5 pixels (being less than 0.36 μm). Proximal foci were between 6 and 15 pixels (0.36–1.07 μm). Distant foci were further than 15 pixels apart (greater than 1.07 μm). These distances were set based on the number of pixels that made up the diameter of an EGFP-SNM1A foci being calculated to be 6 pixels. Examples of these foci at pixel resolution level can be seen in Fig. 2A. The macro script used can be found at: https://doi.org/10.5281/zenodo.11209331.

### Isolation of GST-SNM1A peptides

To establish the role of binding of SNM1A and mutants with PAR chains, PCNA and ubiquitinated PCNA (PCNA[ub]) we generated GST-SNM1A peptide constructs. Consideration was given to the surrounding sequence when designing these peptides – a multiple sequence alignment was used to define the borders of the UBZ, PBZ domains and PIP box (Supplementary Fig. 9). Extra residues both upstream and downstream were included being 2 + 2 in the UBZ/PBZ peptide and 10 + 12 in the PIP box peptide.

The peptide sequences used were (the UBZ domain was mutated at C125F (underlined). The PBZ domain was mutated to give C155A and C161A (underlined)):

RPVYDGYCPNCQMPFSSLIGQTPRWHVFECLDSPPRSE-TECPDGLLCTSTIPFHYKRYTHFLLAQSRAG.

The PIP box peptide was mutated to give Y562A and F563A (underlined):

ARHPSTKVMKQMDIGVYFGLPPKRKEEKLL:

Sequences encoding for these peptides were cloned into the pET-41b vector (Novagen) using the restriction sites MfeI and SalI using standard methods and verified by sequencing.

Plasmids were transformed into competent BL21 E. coli cells by heat shock. Cultures (250 mL) were grown in 2 x YT media at 37 °C until OD 0.6 was reached at which point the incubation temperature was reduced to 16 °C and protein expression was induced with IPTG (0.5 mM). Following 12–14 h incubation cells were pelleted in 50 mL volumes and snap frozen before being stored at − 80 °C.

### Preparation of recombinant PCNA and PCNA[ub]

Human PCNA was produced with a His6-tag in E. coli BL21 and purified by immobilised metal affinity chromatography (IMAC) followed by gel filtration using a Superdex 200 10/300 GL column (Cytiva) and 50 mM HEPES, 200 mM NaCl, 1 mM DTT, 10% glycerol, pH 7.5 as running buffer. Monoubiquitylation of human PCNA was performed in vitro as described before[63], using a mutant UbcH5c (S22R) for conjugation. Following the reaction, PCNA[ub] was purified essentially as described[63] by anion exchange chromatography followed by gel filtration. For the anion exchange chromatography, a Mono Q 5/50 GL 1 mL column (Cytiva) was used. Gel filtration was performed as above.

### PCNA and PCNA[ub] binding assay

GST-peptides were isolated by resuspending a 50 mL E. coli pellet in 7.5 mL of TBS-N (Tris-buffered saline with 0.1% NP-40) and supplemented with 1 x protease inhibitor cocktail tablet (Merck). Cells were sonicated for four rounds of one minute on, one minute off, on ice, to release the cellular contents. Lysates were centrifuged ($20,000 \times g$, 30 min at 4 °C); the cleared lysates were added to pre-blocked glutathione-agarose beads (ThermoScientific, 400 μL of beads per sample, blocked with 5% FCS in PBS for 1 h). Binding was performed at 4 °C for 2 h. Peptide-bound beads were washed with PBS containing various salt concentrations (250 mM, 500 mM, 1 M, 500 mM, 250 mM) before being resuspended in standard PBS. Samples were then divided and binding to PCNA or PCNA[ub] performed. Purified PCNA or PCNA[ub] (150 ng) was added to each sample and tumbled at 4 °C for 1 h. Washes were performed as above to remove unbound PCNA/PCNA[ub]. The washed beads were resuspended in 2 x Laemmli buffer and boiled before being run on 4–12% SDS-PAGE gels and transferred to PVDF membrane for immunoblotting with anti-SNM1A, PCNA, PCNA[ub] and GST.

### PAR Binding assay

To assess the binding of PAR (poly-ADP-ribose) chains to the ligand-binding motifs in SNM1A we utilised the GST-SNM1A PBZ/UBZ peptides described above. Native GST-SNM1A peptides were extracted from BL21 cell lysates (as described for the PCNA binding assay above) and the peptides were eluted from the beads with excess glutathione. The peptides were then dot-blotted to PVDF. The membrane containing the GST-SNM1A peptides as well as BSA and a known PAR interacting purified protein (APLF) as controls.

The membranes were blocked with 10% skimmed milk powder in PBS (10% milk) and a solution of Poly(ADP-ribose) polymers (1:1000 PAR chains in 10% milk, R&D Systems) were incubated at room temperature (1 hr) to allow binding of the PAR chains to the proteins/peptides bound to the membrane. Following extensive washing (3 × 3 min) in TBS-T (Tris-buffered saline solution with 0.1% Tween 20), membranes were probed with an anti-poly-ADP-ribose (1:1000 in 10% milk, Merck) as well as anti-APLF and anti-GST as loading controls.

### Nuclease assay; substrate preparation and assay conditions

Full-length SNM1A and its variants were purified as described[17]. Human EXO1b was purified as described[64] by the Oxford Protein Production Facility, using constructs kindly provided by Paul Modrich. Nuclease assays were performed as described[44]. Oligonucleotide substrates including those containing terminal and internal 8-oxoguanine, thymine glycol, and hypoxanthine lesions were from Eurofins, MWG Operon, Germany. The sequences and features of the oligonucleotides used are described in detail elsewhere[21].

Substrates were prepared as follows: 10 pmol of DNA oligonucleotide (Eurofins MWG Operon, Germany) were 3′-end labelled with 3.3 pmol of α-$^{32}$P-dATP (Perkin Elmer) using terminal deoxynucleotidyl transferase (TdT, 20 U; ThermoFisher Scientific), incubated together at 37 °C for 1 h. Unincorporated nucleotides were removed using a P6 Micro Bio-Spin chromatography column (BioRad). For preparation of double-stranded substrates, radiolabelled single-strand oligonucleotides were annealed with the appropriate unlabelled oligonucleotides (1:1.5 molar ratio of labelled to unlabelled oligonucleotide) by heating

to 95 °C for 5 min, then slowly cooling to room temperature in annealing buffer (10 mM Tris–HCl; pH 7.5, 100 mM NaCl, 0.1 mM EDTA).

Nuclease assays were performed in 10 μL final volume mixtures containing 20 mM HEPES-KOH, pH 7.5, 50 mM KCl, 10 mM $MgCl_2$, 0.05% Triton X-100, 5% glycerol and ~ 20 nM of full-length SNM1A, 1 nM ΔSNM1A or 50 nM EXO1. Reactions were initiated by the addition of DNA (10 or 100 nM), with incubation at 37 °C for the indicated time period. Reactions were terminated by adding 10 μl stop solution (95% formamide, 10 mM EDTA, 0.25% xylene cyanol, 0.25% bromophenol blue) (boiled at 95 °C for 3 min). To examine the effects of PAR chains on SNM1A activity, 5 min pre-incubation with PAR ( + = 100 nM, + + = 1000 nM) or an excess of PARP1 (200 nM) and NAD + (200 μM) were performed.

Reaction products were analysed using 20% denaturing poly-acrylamide gel electrophoresis (40% solution of 19:1 acrylamide:bis-acrylamide, BioRad) containing 7 M urea (Sigma Aldrich) in 1 × TBE (Tris-borate EDTA) buffer at 700 V for 75 min. Gels were fixed for 40 min in a 50% (v/v) methanol, 10% (v/v) acetic acid solution, and vacuum-dried at 80 °C for 2 h. Gels were imaged using a Phosphor Imager screen and scanned using a Typhoon FLA 9500 phosphor-imager (Cytiva Life Sciences).

### SNM1A processing of Zeocin-induced DNA damage in vitro assays

To induce Zeocin damage, DNA plasmids (3.6 μg of pG46) were treated with Zeocin in 1 x reaction buffer[65] (12.5 mM Tris–HCl pH 8.0, 300 mM sucrose, 0.0188% Triton X-100, 1.25 mM EDTA, 5 mM $MgCl_2$ with freshly added 7.5 mM β-mercaptoethanol, 1% heat-inactivated BSA and 100 mM ferrous ammonium sulphate) in 50 μL volumes for 20 min at 37 °C. PCR clean-up columns (Qiagen) were used as per the manu-facturer's instructions to clean up the reacted plasmid DNA and eluted in a 50 μL volume in buffer EB (Qiagen).

The ability of SNM1A to digest Zeocin-induced damaged DNA was measured using a Hind III linearised plasmid (a preferred substrate for SNM1A) by incubating with varying concentrations of SNM1A for 1 h at 37 °C. Following digestion, the samples (3:1) were diluted in stop buffer (95% v/v Formamide, 10 mM EDTA, 0.25% v/v Bromophenol blue) and resolved in 1% agarose gels in 1 x TAE buffer and visualised with ethi-dium bromide. The digestion time course was also performed under the same buffer conditions as above with 1 μM SNM1A.

### Homologous Recombination Repair (HR) and Non-Homologous End-Joining repair (NHEJ) reporter assays

HR and NHEJ repair assays were performed as described[66,67]. Plasmids pCMV-SceI and pDR-GFP (a kind gift from Valentine Macaulay) were used to measure HR and pimEJ5GFP (Addgene # 44026) and pCMV-SceI were used to measure NHEJ[67]. Plasmids (2.6 μg of each) were transfected into $5 \times 10^5$ cells (293FT, 293FT cells treated with siRNA to BRCA2, XRCC4⁻ or SNM1A⁻ 293FT cells) in 2 mL in wells of a 6-well plate, as described above. After 24 h cells were harvested and resuspended in fresh media containing no phenol red and analysed on an Attune NxT Flow Cytometer (Invitrogen) to assess GFP-positive cells as a measure of repair in each pathway.

### Reporting summary

Further information on research design is available in the Nature Portfolio Reporting Summary linked to this article.

## Data availability

All data supporting the findings of this study are available within the paper and its Supplementary Information. Source data are provided in this paper. Previously published PDB depositions specifically men-tioned: 2MRF, 2KQB, 1AXC. Source data are provided with this paper.

## Code availability

The foci macro script used in this study can be found at GitHub (https://github.com/LonnieSwift/macro-script/blob/main/Foci%20Macros)[68].

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

## Acknowledgements

This work was supported by a Cancer Research UK Programme Award (CRUK/ A24759) to P.J.M. and C.J.S. M.R. receives an MRC Graduate Studentship, L.R.H. has received a Wellcome Trust Studentship and S.L. was supported by a National Science Scholarship from the Singaporean Agency for Science, Technology, and Research (A*STAR). We thank Opher Gileadi for purified APLF, Paul Modrich for EXO1 constructs, Philip Hublitz for help with CRISPR guide design, and Andrew Blackford for sharing XRCC4 disrupted cells. George-Lucian Moldovan kindly provided 293 T PCNA$^{K164R}$ cells.

## Author contributions

L.P.S., B.C.L., and D.W. performed cell biology experiments. H.T.B., L.R.H., B.S., S.L., A.C.M. and J.N. undertook protein purification and biochemical assays. M.R. produced the AlphaFold models and figures and undertook data analysis. C.R. and H.D.U. provided PCNA and PCNA$^{ub}$ reagents and expertise. L.P.S. conducted all other experiments. C.J.S. and P.J.M. supervised the project. LPS and PJM wrote the manuscript.

## Competing interests

The authors declare no competing interests.
