## [Peer Review File · Nature Communications]

SNM1A (DCLRE1A) is crucial for efficient repair of complex DNA breaks in human cellsREVIEWER COMMENTS

Reviewer #1 (Remarks to the Author):

The manuscript by Swift et al., describes a new finding on the cellular role of the SNM1A protein of human cells. The authors show that this nuclease is required for repair of lesions generated by zeocin and ionizing radiation. They follow up this observation with biochemical studies of purified SNM1A to show that it can degrade DNA treated with zeocin in vitro, which is resistant to digestion by other nucleases. The manuscript is well written, and the conclusions will be of general interest to the DNA damage and repair community and to cancer biologists. There are, however, some aspects of the manuscript that could be improved.

Specific Remarks:

- 1) The immunofluorescence analysis of SNM1A localization in relation to the site of DSBs (defined by the location of 53BP1) is interesting but seems incomplete. The two proteins localize close to, but not coincident with, one another, which is quite curious. One way to define whether these proteins really co-localize is to perform super-resolution microscopy. Also, are there other DSB markers that could be identified that might perfectly co-localize with SNM1A rather than lie adjacent to it?
- 2) It would be helpful in Figure 3 to indicate how the band labeled as PCNA^{Ub} on the gel was confirmed to be ubiquitylated PCNA.
- 3) The data in Figure 6 are central to the model and yet the biochemistry of the action of SNM1A on the zeocin-treated DNA is not really pursued in sufficient detail. The assay used with plasmid DNA is (at best) semi-quantitative. Couldn't a more quantitative assay be developed – perhaps using an oligonucleotide with a specific chemical modification representative of a DNA end-blocking lesion? Also, the DNA seems to diminish in intensity generally after SNM1A treatment (in panel C) of the zeocin-treated DNA. Why is that? It appears as if the assay was performed using too high a concentration of SNM1A. A titration using lower concentrations might be better. Also, analyzing a time course at a fixed concentration of protein would be useful and informative.

Reviewer #2 (Remarks to the Author):

In their manuscript, Swift et al. describe the exonuclease SNM1A as a new player in repairing DNA double-strand breaks (DSBs). Previous studies had highlighted SNM1A's involvement in processing DNA interstrand crosslink lesions and facilitating break-induced replication of telomeres. The authors extend these findings by revealing SNM1A's contribution to end-joining mediated repair of DSBs.

In particular, they provide evidence that cells lacking SNM1A are more sensitive to ionizing radiation and radiomimetic drugs. A significant aspect of the study is the observation that SNM1A is recruited to DSB sites, a process influenced by specific domains within the SNM1A protein. The authors also investigate an interaction between SNM1A and PCNA, in vitro and in cells. Additionally, they conduct detailed biochemical analyses to understand SNM1A's ability to process complex, non-blunt DNA breaks.

DSBs are generally highly complex lesions that require a range of enzymes to prepare them for repair. The mechanics of this process at the molecular level are a central focus in the DSB repair field. In this context, Swift and colleagues' manuscript makes a timely and pertinent contribution. Its implications also extend beyond genome stability to areas like cancer- and radiobiology, adding additional relevance to the findings presented in this manuscript.

The manuscript is well written, its structure is coherent, and its scientific approach is overall sound.

However, there are several issues that dampen my enthusiasm for the manuscript in its current form. Chief among these concerns is the authors' propensity to overinterpret data, leading to the formulation of a model for SNM1A's contribution to end-joining-mediated repair. Presently, several pivotal conclusions that are essential for this model appear to be derived solely from disparities in recruitment kinetics. To bolster their claims, the authors should provide more direct evidence (beyond mere inference from recruitment kinetics) for RAD18 being the E3 ligase that mediates the UBZ/ubiquitin-dependent recruitment of SNM1A to breaks, for PCNA's position upstream of SNM1A, and for K164 as the ubiquitylated residue that is recognized by SNM1A at DSBs.

In addition, I am surprised that the authors place SNM1A in such a firm way on the NHEJ side of DSB repair. This assertion seems to disregard prior findings, including those suggesting PCNA's involvement in D-loop extension during homologous recombination (HR) (<https://doi.org/10.1016/j.dnarep.2013.05.001>), as well as its role in promoting Exo1-mediated DNA end resection at DSBs (<https://doi.org/10.1093/nar/gkt672>). To solidify SNM1A's place within the NHEJ process, further experiments are needed to explicitly exclude any potential role for SNM1A in HR.

Specific comments:

- Regarding Figure 1D and E: Does complementation also rescue sensitivity to IR in U2OS cells and Zeocin sensitivity in HEK293FT cells?
- Concerning Figure 1F/ I: Could the authors clarify the method used for foci quantification? The quantification in Figure 1F suggests the presence of individual foci. However, I have difficulty to discern individual foci and would like to see a clarification of what defines a focus in this dataset.
- I feel that the wording "profound" sensitivity is quite strong given that the sensitivity to IR is only really apparent at higher doses and that γ H2AX/53BP1 foci are monitored at 0.1mg/ml Zeocin, which also is a very high dose.
- In Figure 1I, the representative images appear to deviate somewhat from the implications of the quantification. While the images suggest a substantial persistence of γ H2AX at 24h in contrast to control cells, the quantified data do not corroborate this observation. This discrepancy raises questions about the representativeness of the images showcased in Figure 1I. A related point of clarification pertains to the experimental conditions. It's crucial to elucidate whether Zeocin application followed a specific duration, succeeded by recovery phases of 2 and 24 hours, or if Zeocin was consistently administered throughout the experiment. In the latter scenario, drawing conclusions about the "persistence of DNA lesions" becomes challenging, given that lesions are continually induced.
- I'm surprised that SNM1A-deficient cells are not sensitive at all to H₂O₂, given that H₂O₂ also induces ROS-mediated DSBs. It would be worth to discuss this briefly in the text.
- Figure 2A: The authors state in the text that "the majority of 53BP1 foci colocalize with EGFP-SNM1A foci". Again, this is a massive overstatement and is just not in the data that is presented. Figure 2A shows 1 cell and 5 insets and I don't see much colocalization. Proximity, yes, but not colocalization. The authors need to quantify this experiment to make such a strong statement. Also, what is the nature of the EGFP-SNM1A foci that are not in proximity to 53BP1? And what about the endogenous protein - does it form damage-induced foci or accumulate at laserlines?
- I am not convinced that the SNM1A foci shown in Figure 2A are specific to Zeocin treatment. They could also be an overexpression artifact. This possibility is supported by Figure 4A and the G2 pre-stripe image in Figure 4D. Is this something the authors have considered?
- Figure 3: The quality disparity between microscopy in Figure 3 and Figure 2 is quite conspicuous. I find it again extremely challenging to discern individual foci for EGFP-SNM1A in Figure 3E and F, let alone verify colocalization with 53BP1 as suggested in the text. Overall, I find the quantification of these images quite puzzling as it does not match with what I see in the images – this is particularly true for the PBZ and PIP domain constructs. Also, several images are out of focus – this needs to be remedied to produce an acceptable dataset. Higher image resolution would also improve overall figure quality.
- Also regarding Figure 3, is EGFP-SNM1A focus formation correlated to expression levels? If this is the

case, the authors need to control for similar expression levels between mutants.

- Regarding Figure 4, there is again a disconnect in image quality between Panel A and B/D. The laser line data in A is convincing, but what cell type is this in, which timepoint is shown etc? This lack of experimental detail is, unfortunately, pervasive throughout the manuscript and makes the data oftentimes difficult to evaluate.

- I'm confused by how the authors reliably determined cell cycle stage in Figure 4. They state in the methods section that they did this with PCNA foci as S-phase marker and DAPI. In the actual image in Figure 4D, I see neither marker. Also, how would these approaches work in live cell imaging? I presume they used RFP-PCNA foci, but DAPI would have been done in fixed cells.

- Regarding Figure 5: The image quality here is very poor as many nuclei appear to be again out of focus. Is this the same cell type as in Figure 4?

- The authors conclude that PCNA has a "key role" in recruiting SNM1A to DSBs. This is based primarily on the recruitment kinetics of the two proteins. However, the authors need to flesh this out a bit more. What happens to SNM1A recruitment in the absence of PCNA?

- RAD18 is downstream of MDC1/ RNF8 in the DSB response (<https://doi.org/10.1038/ncb1865>). It would be helpful if the authors discussed possibilities as to why RNF8 (and RNF168) do then not affect SNM1A recruitment. Also, is RAD18 E3 ligase activity required? Is PARP required? Are apical kinases such as DNA-PK and ATM involved in SNM1A recruitment?

- To delve into possible roles in HR, the manuscript should additionally address whether SNM1A contributes to PCNA-dependent EXO1 recruitment, whether it influences end-resection, and - related to this- whether it affects RPA/RAD51 in the context of SNM1A depletion.

Reviewer #3 (Remarks to the Author):

In this manuscript, Swift et al describe a novel function for the nuclease SNM1A in the processing of "dirty" DSBs. This observation is of general interest for the DNA repair field and the readers of Nature Communications. Some critical experiments are however required to non-ambiguously demonstrate such function.

The authors first show that: 1) SNM1A knockout cells are hyper-sensitive to zeocin and to irradiation. 2) They see an accumulation of unrepaired DSBs (persistence of gH2A.X and 53BP1 foci) in SNM1A-cells, as well as accumulation of cells in G2. 3) They see that SNM1A is recruited in the proximity of 53BP1 foci. And finally 4) they found no reduced activity of HR or NHEJ when using nucleases that generate clean DSBs. The authors conclude from these data that SNM1A is required for the repair of complex ("dirty") DSBs.

1. As noted by the authors, SNM1A was shown to be essential during ALT, but all of these data are produced in ALT+ cells, with the exception of Figure 1E which shows that SNM1A- HEK293 are also more sensitive to zeocin, though not remotely to the extent of ALT+ U2OS cells. To confirm that the study is not specific to DSB repair in ALT cells, authors should reproduce the cell cycle analysis and foci accumulation in HEK293.

2. They need to show that the recruitment of SNM1A is specific to dirty DSBs, by performing proper quantification of the percentage of 53BP1 foci that have SNM1A co-localized (or in the vicinity) upon induction of clean vs dirty damage: Zeocin (dirty), IR (mix) and a nuclease such as AsiSI (clean, where SNM1A should not be recruited).

The authors then focus on the motifs of SNM1A that promote its recruitment to DSBs. They find that the PAR-binding, PIP-motif, and E3-Ubiquitin binding cooperate for the recruitment and retention of SNM1A to DSBs, but do not promote its activity. The laser-irradiation experiments are well-performed, and the data is convincing. However, the data using zeocin needs to be better analyzed:

3. Data figure 3G should represent the number of SNM1A/gH2A.X and SNM1A/53BP1 colocalization. Of note, the images shown are not representative of the current quantification (for instance, PBZ mutant is highly affected in quantification but not in IF image, and vice-versa for PIP mutant).

Finally, the authors use in vitro experiments to show that SNM1A nuclease activity remains active on DNA substrate with various lesions, including on DSBs generated with zeocin treatment.

4. This is a first step, but to definitively conclude that SNM1A promote the repair of dirty DSBs, they need to show in cells that HR and/or NHEJ requires SNM1A at complex DSBs. One way to approach that would be to in vitro digest and modify the ends of the HR and NHEJ reporters, and transfect them into SNM1A+ or KO cells. If SNM1A indeed promotes the repair of complex DSBs, authors should find no HR/NHEJ reduction in SNM1A- cells when the DSB ends have not been modified, but a reduced activity at modified ends.

REVIEWER COMMENTS

Reviewer #1 (Remarks to the Author):

The manuscript by Swift et al., describes a new finding on the cellular role of the SNM1A protein of human cells. The authors show that this nuclease is required for repair of lesions generated by zeocin and ionizing radiation. They follow up this observation with biochemical studies of purified SNM1A to show that it can degrade DNA treated with zeocin in vitro, which is resistant to digestion by other nucleases. The manuscript is well written, and the conclusions will be of general interest to the DNA damage and repair community and to cancer biologists. There are, however, some aspects of the manuscript that could be improved.

We thank the reviewer for their careful reading of the manuscript and their constructive comments.

Specific Remarks:

1) The immunofluorescence analysis of SNM1A localization in relation to the site of DSBs (defined by the location of 53BP1) is interesting but seems incomplete. The two proteins localize close to, but not coincident with, one another, which is quite curious. One way to define whether these proteins really co-localize is to perform super-resolution microscopy. Also, are there other DSB markers that could be identified that might perfectly co-localize with SNM1A rather than lie adjacent to it?

The co-proximity, or adjacency of the SNM1A foci to the regions marked by 53BP1 (and γ HA2X) is interesting, as the reviewer states. However, this is not surprising or unprecedented. Going back to the pioneering work of the Lukas lab, (Bakker-Jensen et al, JCB, 173, P 195, 2006) it is clear that the chromatin-associated markers associated with DSBs (γ H2AX, 53BP1) all occupy megabase-long 'nanodomains' that are colocalised. By contrast repair factors (BRCA2, FANCD2, etc) accumulate in distinct subchromatic microcompartments associated with DNA processing. We would therefore argue that the pattern of staining we see would – if anything – be expected, since SNM1A is a repair/processing enzyme and would be expected to be excluded from the chromatin domains typified by 53BP1 accumulation. Nonetheless, we are conscious that some of the wording employed in the original submission implies colocalisation, and we have been careful to rephrase this, and to also add discussion around the pattern of organisation of the SNM1A and 53BP1 foci in accordance with the (supportive and consistent) literature (at the start of the Discussion).

2) It would be helpful in Figure 3 to indicate how the band labeled as PCNA^{ub} on the gel was confirmed to be ubiquitylated PCNA.

It can be seen from visual inspection of Fig. 3C, left hand panel, that the recombinant form of PCNA^{ub} is almost 100% modified by the expected band shift, where the membrane was probed with a pan-PCNA antibody. We have added a new panel to Fig. 3, right hand panels, where probing with either the pan-PCNA antibody or a high selective anti PCNA^{ub} antibody (originally derived and carefully validated by Roger Woodgate's laboratory) confirm that only the ubiquitinated form of PCNA binds to the UBZ-containing peptide.

3) The data in Figure 6 are central to the model and yet the biochemistry of the action of SNM1A on the zeocin-treated DNA is not really pursued in sufficient detail. The assay used with plasmid DNA is (at best) semi-quantitative. Couldn't a more quantitative assay be developed – perhaps using an

oligonucleotide with a specific chemical modification representative of a DNA end-blocking lesion? Also, the DNA seems to diminish in intensity generally after SNM1A treatment (in panel C) of the zeocin-treated DNA. Why is that? It appears as if the assay was performed using too high a concentration of SNM1A. A titration using lower concentrations might be better. Also, analyzing a time course at a fixed concentration of protein would be useful and informative.

Panel 6A shows an oligo with an end blocking (Exo1 blocking) lesion, which SNM1A is able to readily digest. The reason the intensity of DNA diminishes during SNM1A digestion in panel 6C is that the DNA goes from being a discrete molecule of a specific size to being digested into ever smaller fragments in a step-wise manner and the concentration of each of these products is low as a function of the total DNA present. The gel is stained with SYBR Gold to allow direct visualisation of all the product molecules present. In panel C, lanes containing HindIII linearised DNA were used to ensure that the concentrations of enzyme employed permit analysis of the digestion pattern across a suitable range of concentrations. We have performed a time-course and this forms new Suppl. Fig. 11F and 11G.

Reviewer #2 (Remarks to the Author):

In their manuscript, Swift et al. describe the exonuclease SNM1A as a new player in repairing DNA double-strand breaks (DSBs). Previous studies had highlighted SNM1A's involvement in processing DNA interstrand crosslink lesions and facilitating break-induced replication of telomeres. The authors extend these findings by revealing SNM1A's contribution to end-joining mediated repair of DSBs.

In particular, they provide evidence that cells lacking SNM1A are more sensitive to ionizing radiation and radiomimetic drugs. A significant aspect of the study is the observation that SNM1A is recruited to DSB sites, a process influenced by specific domains within the SNM1A protein. The authors also investigate an interaction between SNM1A and PCNA, in vitro and in cells. Additionally, they conduct detailed biochemical analyses to understand SNM1A's ability to process complex, non-blunt DNA breaks.

DSBs are generally highly complex lesions that require a range of enzymes to prepare them for repair. The mechanics of this process at the molecular level are a central focus in the DSB repair field. In this context, Swift and colleagues' manuscript makes a timely and pertinent contribution. Its implications also extend beyond genome stability to areas like cancer- and radiobiology, adding additional relevance to the findings presented in this manuscript.

The manuscript is well written, its structure is coherent, and its scientific approach is overall sound. However, there are several issues that dampen my enthusiasm for the manuscript in its current form. Chief among these concerns is the authors' propensity to overinterpret data, leading to the formulation of a model for SNM1A's contribution to end-joining-mediated repair. Presently, several pivotal conclusions that are essential for this model appear to be derived solely from disparities in recruitment kinetics. To bolster their claims, the authors should provide more direct evidence (beyond mere inference from recruitment kinetics) for RAD18 being the E3 ligase that mediates the UBZ/ ubiquitin-dependent recruitment of SNM1A to breaks, for PCNA's position upstream of SNM1A, and for K164 as the ubiquitylated residue that is recognized by SNM1A at DSBs.

In addition, I am surprised that the authors place SNM1A in such a firm way on the NHEJ side of DSB repair. This assertion seems to disregard prior findings, including those suggesting PCNA's

involvement in D-loop extension during homologous recombination (HR) (<https://doi.org/10.1016/j.dnarep.2013.05.001>), as well as its role in promoting Exo1-mediated DNA end resection at DSBs (<https://doi.org/10.1093/nar/gkt672>). To solidify SNM1A's place within the NHEJ process, further experiments are needed to explicitly exclude any potential role for SNM1A in HR.

We thank the reviewer for their careful reading of the manuscript and their constructive comments. We do not have a heavy bias towards the idea repair mainly goes through the NHEJ pathway and have been careful not imply as such in the manuscript. We were careful not to pin down the role of SNM1A to a specific subpathway of DSB repair as clearly it might contribute to several.

Specific comments:

- Regarding Figure 1D and E: Does complementation also rescue sensitivity to IR in U2OS cells and Zeocin sensitivity in HEK293FT cells?

Yes, we have shown complementation to the 293FT cells and have added this as new panel 1E.

- Concerning Figure 1F/ I: Could the authors clarify the method used for foci quantification? The quantification in Figure 1F suggests the presence of individual foci. However, I have difficulty to discern individual foci and would like to see a clarification of what defines a focus in this dataset.

We apologise that the rendering of these images was not ideal in the original submission. We have added an improved version of these images (Fig. 1I) and also ensure that the method used to identify and define a focus are completely described in the Materials and Methods section. We have also added a new Suppl. Figs 3 and 5 that shows a wider field of cells to reassure that we were in no way 'cherry picking' the cells for visual analysis. Moreover, the scoring algorithm for focus analysis is automated and therefore effectively blinded to the identity of the sample.

- I feel that the wording "profound" sensitivity is quite strong given that the sensitivity to IR is only really apparent at higher doses and that γ H2AX/53BP1 foci are monitored at 0.1mg/ml Zeocin, which also is a very high dose.

We agree, thanks. We have removed this wording throughout.

- In Figure 1I, the representative images appear to deviate somewhat from the implications of the quantification. While the images suggest a substantial persistence of γ H2AX at 24h in contrast to control cells, the quantified data do not corroborate this observation. This discrepancy raises questions about the representativeness of the images showcased in Figure 1I. A related point of clarification pertains to the experimental conditions. It's crucial to elucidate whether Zeocin application followed a specific duration, succeeded by recovery phases of 2 and 24 hours, or if Zeocin was consistently administered throughout the experiment. In the latter scenario, drawing conclusions about the "persistence of DNA lesions" becomes challenging, given that lesions are continually induced.

Please see above – we agree the rendering of these images was not ideal in the original submission. We have added an improved version of these images (Fig. 1I) We have also added a new Suppl. Fig. 3 that shows a wider field of cells to reassure that we were in no way ‘cherry picking’ the cells for visual analysis. Moreover, focus scoring is performed by a macro script and is therefore essentially ‘blinded’ or subject to investigator bias. We have made this point clearer in the Materials and Methods.

The Zeocin treatment is ‘hit and run’ (2 hours, washed away and then recovery for up to 24 hours) to ensure recovery time represents repair of damage induced in an acute manner.

- I’m surprised that SNM1A-deficient cells are not sensitive at all to H₂O₂, given that H₂O₂ also induces ROS-mediated DSBs. It would be worth to discuss this briefly in the text.

We think this is because, while H₂O₂ is capable of producing DSBs, they are infrequent compared to the single-strand breaks and oxidised bases that are the overwhelming majority of damage that must be dealt with, mainly by BER enzymes. We have briefly commented.

- Figure 2A: The authors state in the text that “the majority of 53BP1 foci colocalize with EGFP-SNM1A foci”. Again, this is a massive overstatement and is just not in the data that is presented. Figure 2A shows 1 cell and 5 insets and I don’t see much colocalization. Proximity, yes, but not colocalization. The authors need to quantify this experiment to make such a strong statement. Also, what is the nature of the EGFP-SNM1A foci that are not in proximity to 53BP1? And what about the endogenous protein - does it form damage-induced foci or accumulate at laserlines?

The co-proximity, or adjacency of the SNM1A foci to the regions marked by 53BP1 (and γ HA2X) is interesting, as the reviewer states. However, this is not surprising or unprecedented. Going back to the pioneering work of the Lukas lab, (Bakker-Jensen et al, JCB, 173, P 195, 2006) it is clear that the chromatin-associated markers associated with DSBs (γ H2AX, 53BP1) all occupy megabase-long ‘nanodomains’ that are colocalised. By contrast repair factors (BRCA2, FANCD2, etc) accumulate in distinct subchromatic microcompartments associated with DNA processing. We would therefore argue that the pattern of staining we see would – if anything – be expected, since SNM1A is a repair/processing enzyme and would be expected to be excluded from the chromatin domains typified by 53BP1 accumulation. Nonetheless, the reviewer is quite right that we implied colocalization of SNM1A and 53BP1 in the original submission, and been careful to amend to proximity/adjacency. We have also added discussion around the pattern of organisation of the SNM1A and 53BP1 foci in accordance with the (supportive and consistent) literature.

The endogenous protein has been impossible to study to date. We have raised multiple poly- and mono-clonal antibodies to SNM1A, in addition to trialling every commercially available antibody, but to no avail. We have also attempted to tag endogenous SNM1A but this failed, despite us having success with other DNA repair targets. One thing is clear -- the endogenous protein is expressed at low levels.

- I am not convinced that the SNM1A foci shown in Figure 2A are specific to Zeocin treatment. They could also be an overexpression artifact. This possibility is supported by Figure 4A and the G2 pre-stripe image in Figure 4D. Is this something the authors have considered?

EGFP-SNM1A forms spontaneous foci during S-phase coincident with PCNA. Indeed, (endogenous) SNM1A has been found as a component of the replisome in multiple iPOND-MS studies. The cells

that show SNM1A foci, for example Fig. 4, cited, are in S-phase. In new Suppl. Fig. 5 we show a wider field of cell that demonstrate show a few (S-phase cells) in any field show SNM1A foci, but that the induction upon Zeocin treatment is marked.

- Figure 3: The quality disparity between microscopy in Figure 3 and Figure 2 is quite conspicuous. I find it again extremely challenging to discern individual foci for EGFP-SNM1A in Figure 3E and F, let alone verify colocalization with 53BP1 as suggested in the text. Overall, I find the quantification of these images quite puzzling as it does not match with what I see in the images – this is particularly true for the PBZ and PIP domain constructs. Also, several images are out of focus – this needs to be remedied to produce an acceptable dataset. Higher image resolution would also improve overall figure quality.

We agree and apologise – this is primarily a rendering issue when we placed the images and they have not reproduced well. We have addressed this with new versions of the panels, and in some cases replaced the panels with better quality images. Moreover, as is evident from Fig. 3G, there is a lot of noise in the system, and this is precisely the reason we moved to the quantifiable laser stripe method.

- Also regarding Figure 3, is EGFP-SNM1A focus formation correlated to expression levels? If this is the case, the authors need to control for similar expression levels between mutants.

This is not an issue – we carefully assessed the baseline EGFP signal to ensure it was equal of each cell prior to striping to avoid this pitfall.

- Regarding Figure 4, there is again a disconnect in image quality between Panel A and B/D. The laser line data in A is convincing, but what cell type is this in, which timepoint is shown etc? This lack of experimental detail is, unfortunately, pervasive throughout the manuscript and makes the data oftentimes difficult to evaluate.

Again, the images were not well rendered and have been improved. We have been careful to ensure that figures are fully annotated and the legends fully descriptive in the revision and apologise for the omission of this information in several places in the original submission.

- I'm confused by how the authors reliably determined cell cycle stage in Figure 4. They state in the methods section that they did this with PCNA foci as S-phase marker and DAPI. In the actual image in Figure 4D, I see neither marker. Also, how would these approaches work in live cell imaging? I presume they used RFP-PCNA foci, but DAPI would have been done in fixed cells.

We have added the details to the methods with a clearer explanation: Essentially S-phase cell are indeed those with PCNA foci. G2 cells can be readily identified by their rounded-up phenotype and 'flat' PCNA negative cells are in G1 phase.

- Regarding Figure 5: The image quality here is very poor as many nuclei appear to be again out of focus. Is this the same cell type as in Figure 4?

Yes, agree, we improved the rendering of these panels. We have added cell type to the figure legend.

- The authors conclude that PCNA has a “key role” in recruiting SNM1A to DSBs. This is based primarily on the recruitment kinetics of the two proteins. However, the authors need to flesh this out a bit more. What happens to SNM1A recruitment in the absence of PCNA?

PCNA cannot be removed or depleted – this is lethal, even for a short time.

- RAD18 is downstream of MDC1/ RNF8 in the DSB response (<https://doi.org/10.1038/ncb1865>). It would be helpful if the authors discussed possibilities as to why RNF8 (and RNF168) do then not affect SNM1A recruitment. Also, is RAD18 E3 ligase activity required? Is PARP required? Are apical kinases such as DNA-PK and ATM involved in SNMA1 recruitment?

In the context of its role in PCNA monoubiquitination (rather than SLF1-2 signalling) RNF8 and RNF168 are dispensable for RAD18 recruitment/activity (Raschle et al, Science. 2015 May 1; 348(6234): 1253671 (ref 51)), consistent with our results, and we discuss this point on page 11.

We believe we have done plenty to establish that ubiquitination of PCNA residue K164 by RAD18 is required for SNM1A recruitment. We have shown direct interaction between the UBZ and PCNA^{ub}, demonstrated that RAD18 is required for SNM1A recruitment to laser stripes, and that substitution of the single lysine residue (K164) that is uniquely modified by RAD18 is required for efficient SNM1A recruitment. We have done preliminary experiments with ATM, ATR and DNA-PKcs inhibitors and saw no major effects. That is not to say that subtle defects might be present, but these kinases do not appear to be essential for SNM1A recruitment. Due the fact that there could be subtle roles for these PIKKS we have avoided discussing in this manuscript as we would not want to mislead those studying further fine details of this process in the future.

- To delve into possible roles in HR, the manuscript should additionally address whether SNM1A contributes to PCNA-dependent EXO1 recruitment, whether it influences end-resection, and - related to this- whether it affects RPA/RAD51 in the context of SNM1A depletion.

As we mentioned previously, at this stage we have deliberately avoided implicating SNM1A in specific DSB pathways at it is plausible it contributes to several pathways and sorting this out will be a new phase of research.

Reviewer #3 (Remarks to the Author):

In this manuscript, Swift et al describe a novel function for the nuclease SNM1A in the processing of “dirty” DSBs. This observation is of general interest for the DNA repair field and the readers of Nature Communications. Some critical experiments are however required to non-ambiguously demonstrate such function.

We thank the reviewer for their careful reading of the manuscript and their constructive comments.

The authors first show that: 1) SNM1A knockout cells are hyper-sensitive to zeocin and to irradiation. 2) They see an accumulation of unrepaired DSBs (persistence of gH2A.X and 53BP1 foci) in SNM1A-cells, as well as accumulation of cells in G2. 3) They see that SNM1A is recruited in the proximity of 53BP1 foci. And finally 4) they found no reduced activity of HR or NHEJ when using nucleases that generate clean DSBs. The authors conclude from these data that SNM1A is required for the repair of

complex (“dirty”) DSBs.

1. As noted by the authors, SNM1A was shown to be essential during ALT, but all of these data are produced in ALT+ cells, with the exception of Figure 1E which shows that SNM1A- HEK293 are also more sensitive to zeocin, though not remotely to the extent of ALT+ U2OS cells. To confirm that the study is not specific to DSB repair in ALT cells, authors should reproduce the cell cycle analysis and foci accumulation in HEK293.

We agree that these cell lines warrant further investigation. We have performed FACS analysis following Zeocin treatment, shown in new Suppl. Fig. 4. We have also undertaken complementation of the 293FT cells with EGFP-SNM1A constructs to demonstrate this restores Zeocin resistance, ensuring that the phenotype observed is directly due to SNM1A loss in these cell lines (new Fig. 1E).

2. They need to show that the recruitment of SNM1A is specific to dirty DSBs, by performing proper quantification of the percentage of 53BP1 foci that have SNM1A co-localized (or in the vicinity) upon induction of clean vs dirty damage: Zeocin (dirty), IR (mix) and a nuclease such as AsiSI (clean, where SNM1A should not be recruited).

Thank you for the suggestion. We have performed the AsiSi experiment and observed that that SNM1A is in fact recruited to some clean DSBs (new Fig. 2B). We discuss the reasons for this in a modified section of text related to the figure. We have also examined SNM1A focus recruitment of IR induced damage and confirm that this occurs in a manner that is proximal to γ H2AX foci (new Suppl. Fig. 5).

The authors then focus on the motifs of SNM1A that promote its recruitment to DSBs. They find that the PAR-binding, PIP-motif, and E3-Ubiquitin binding cooperate for the recruitment and retention of SNM1A to DSBs, but do not promote its activity. The laser-irradiation experiments are well-performed, and the data is convincing. However, the data using zeocin needs to be better analyzed:

3. Data figure 3G should represent the number of SNM1A/ γ H2A.X and SNM1A/53BP1 colocalization. Of note, the images shown are not representative of the current quantification (for instance, PBZ mutant is highly affected in quantification but not in IF image, and vice-versa for PIP mutant).

We agree and apologise – much of this is primarily a rendering issue when we placed the images and they have not reproduced well. We have addressed this with new versions of the panels, and in some cases replaced the panels with better quality images.

Finally, the authors use in vitro experiments to show that SNM1A nuclease activity remains active on DNA substrate with various lesions, including on DSBs generated with zeocin treatment.

4. This is a first step, but to definitively conclude that SNM1A promote the repair of dirty DSBs, they need to show in cells that HR and/or NHEJ requires SNM1A at complex DSBs. One way to approach that would be to in vitro digest and modify the ends of the HR and NHEJ reporters, and transfect them into SNM1A+ or KO cells. If SNM1A indeed promotes the repair of complex DSBs, authors should find no HR/NHEJ reduction in SNM1A- cells when the DSB ends have not been modified, but a reduced activity at modified ends.

This is a good suggestion thanks. We find, however, that the treatment of these reporters reduces the (already) very low efficiency of repair even in WT cell in the assays precluding further analysis. We have outlined future steps that might be used to interrogate SNM1A role in subpathways of DSB repair, but this is clearly a complex question that will form the basis of future studies.

REVIEWER COMMENTS

Reviewer #1 (Remarks to the Author):

I feel that the authors have improved the manuscript and I now recommend that the article be accepted for publication.

Reviewer #2 (Remarks to the Author):

I am quite satisfied with the revised manuscript. The authors have addressed nearly all of my primary concerns, especially those related to the quality of images and the details of the experiments. They have also clarified many aspects that were ambiguous to me upon reading the initial version of the manuscript.

However, there are some points I wish to revisit:

1. „The co-proximity, or adjacency of the SNM1A foci to the regions marked by 53BP1 (and γ H2AX) is interesting, as the reviewer states. However, this is not surprising or unprecedented. Going back to the pioneering work of the Lukas lab, (Bakker-Jensen et al, JCB, 173, P 195, 2006) it is clear that the chromatin-associated markers associated with DSBs (γ H2AX, 53BP1) all occupy megabase-long 'nanodomains' that are colocalised. By contrast repair factors (BRCA2, FANCD2, etc) accumulate in distinct subchromatic microcompartments associated with DNA processing. We would therefore argue that the pattern of staining we see would – if anything – be expected, since SNM1A is a repair/processing enzyme and would be expected to be excluded from the chromatin domains typified by 53BP1 accumulation. Nonetheless, the reviewer is quite right that we implied colocalization of SNM1A and 53BP1 in the original submission, and been careful to amend to proximity/adjacency. We have also added discussion around the pattern of organisation of the SNM1A and 53BP1 foci in accordance with the (supportive and consistent) literature.”

I disagree with this line of arguments (which is also part of the revised discussion section). First, to my knowledge, there is no data showing that 53BP1 is excluded from the actual break site (at least in the context of NHEJ/ G1). On the contrary, one of the functions of 53BP1 (albeit not well understood) is to keep DNA ends in proximity to facilitate end-joining. Also, how would it repress end-resection via Shieldin (another key function) if it is not localised at the break, particularly during early stages of lesion recognition and processing? Second, Simon Bekker-Jensen showed in the cited paper that the "microcompartments", which Swift et al refer to here, are consistent with ssDNA and that they can only be observed in G2 –strongly arguing for HR in this context. While I recognize the authors' effort to avoid categorically assigning SNM1A to a specific repair pathway, the assertion that repair and DNA processing factors generally exclude 53BP1 is inaccurate, as this is neither demonstrated nor claimed in the 2006 Bekker-Jensen study. I'd also like to point to another excellent 2019 paper, also by the Lukas lab, that nicely picks up on this topic using super-resolution imaging: <https://doi.org/10.1038/s41586-019-1659-4>. Here again is no evidence that repair factors are precluded from the 53BP1 domain, at least in the context of NHEJ. What they show is actually that 53BP1 co-localizes with (and surrounds) DSBs that are positive for the NHEJ-repair factor XRCC4 but are mutually exclusive with RPA (and by extension end-resected ssDNA and HR).

2. In light of this, I still think the data in Figure 2A would benefit from some sort of quantitation to give a better idea of how many SNM1A foci are in proximity to 53BP1 vs how many (partially) overlap with 53BP1.

3. "Also regarding Figure 3, is EGFP-SNM1A focus formation correlated to expression levels? If this is

the case, the authors need to control for similar expression levels between mutants.

Rebuttal: "This is not an issue – we carefully assessed the baseline EGFP signal to ensure it was equal of each cell prior to striping to avoid this pitfall."

I meant the foci experiments in Figure 3 (not the laserline (striping) experiments in Figure 4). The authors need to show that all constructs were expressed at similar levels in order to be able to compare their focus forming properties. I find the sentence „we carefully assessed the baseline EGFP signal“ very dissatisfying. How was this assessed? There is no explanation or figure as far as I could tell.

Reviewer #3 (Remarks to the Author):

The revised version of Swift et al. manuscript is mostly similar to its original version, and I remain convinced that the data provided are insufficient to support the authors' claim that NSM1A promotes the repair of dirty ends. I therefore do not support publication in Nature Communications.

1- The authors did not answer my point #1 and did not provide foci accumulation analyses in a different cell line.

2- The AsiSI experiment needs to be quantified. Stating that there is "a degree of proximity" between SNM1A and DSBs markers without any quantification is rather appalling. Since there is recruitment of SNM1A at clean DSBs, which is clearly in opposition to the authors' hypothesis, it is CRITICAL to show that 1) cell survival is unchanged and 2) 53BP1 and gH2A.X foci do not accumulate longer at AsiSI DSBs in SNM1A KO cells compared to WT cells – basically perform the analyses done in Figures 1 A-H but using AsiSI as inducer of DSBs.

3- I asked for a simple quantification for Figure 3 and even that was not done. The manuscript is hugely lacking rigor regarding co-localization analyses.

4- My point #4 was not addressed either.

REVIEWER COMMENTS

Reviewer #1 (Remarks to the Author):

I feel that the authors have improved the manuscript and I now recommend that the article be accepted for publication.

We thank the reviewer for their help in improving the MS.

Reviewer #2 (Remarks to the Author):

I am quite satisfied with the revised manuscript. The authors have addressed nearly all of my primary concerns, especially those related to the quality of images and the details of the experiments. They have also clarified many aspects that were ambiguous to me upon reading the initial version of the manuscript.

However, there are some points I wish to revisit:

1. „The co-proximity, or adjacency of the SNM1A foci to the regions marked by 53BP1 (and γ H2AX) is interesting, as the reviewer states. However, this is not surprising or unprecedented. Going back to the pioneering work of the Lukas lab, (Bakker-Jensen et al, JCB, 173, P 195, 2006) it is clear that the chromatin-associated markers associated with DSBs (γ H2AX, 53BP1) all occupy megabase-long ‘nanodomains’ that are colocalised. By contrast repair factors (BRCA2, FANCD2, etc) accumulate in distinct subchromatic microcompartments associated with DNA processing. We would therefore argue that the pattern of staining we see would – if anything – be expected, since SNM1A is a repair/processing enzyme and would be expected to be excluded from the chromatin domains typified by 53BP1 accumulation. Nonetheless, the reviewer is quite right that we implied colocalization of SNM1A and 53BP1 in the original submission, and been careful to amend to proximity/adjacency. We have also added discussion around the pattern of organisation of the SNM1A and 53BP1 foci in accordance with the (supportive and consistent) literature.”

I disagree with this line of arguments (which is also part of the revised discussion section). First, to my knowledge, there is no data showing that 53BP1 is excluded from the actual break site (at least in the context of NHEJ/ G1). On the contrary, one of the functions of 53BP1 (albeit not well understood) is to keep DNA ends in proximity to facilitate end-joining. Also, how would it repress end-resection via Shieldin (another key function) if it is not localised at the break, particularly during early stages of lesion recognition and processing? Second, Simon Bekker-Jensen showed in the cited paper that the “microcompartments”, which Swift et al refer to here, are consistent with ssDNA and that they can only be observed in G2 –strongly arguing for HR in this context. While I recognize the authors' effort to avoid categorically assigning SNM1A to a specific repair pathway, the assertion that repair and DNA processing factors generally exclude 53BP1 is inaccurate, as this is neither demonstrated nor claimed in the 2006 Bekker-Jensen study. I'd also like to point to another excellent 2019 paper, also by the Lukas lab, that nicely picks up on this topic using super-resolution imaging: <https://doi.org/10.1038/s41586-019-1659-4>. Here again is no evidence that repair factors are precluded from the 53BP1 domain, at least in the context of NHEJ. What they show is actually

that 53BP1 co-localizes with (and surrounds) DSBs that are positive for the NHEJ-repair factor XRCC4 but are mutually exclusive with RPA (and by extension end-resected ssDNA and HR).

We appreciate the helpful dialogue on this point. We agree the cited paper is excellent and agree with the reviewer's assessment. At the same time, it is also clear that shieldin and 53BP1 localise as (at least) partly separate entities, where 53BP1-binding protein RIF1 bridges interactions between ATM-phosphorylated motifs in 53BP1 domains and the SHLD3 subunit of shieldin (DOI: 10.15252/embr.201847560). This represses extensive resection, but might promote fill-in synthesis as part of this process, and it is possible that SNM1A is involved in this, or related events, during complex DSB processing for NHEJ where the spatial relationship if the factors is yet to be worked out. Moreover, for complex breaks there is evidence of limited end processing-related resection occurring in G1 cells (DOI: 10.1016/j.molcel.2016.12.016), but its spatial regulation is not understood. We agree that it seems clear that the details of the nature if the 53BP overlapping and proximal SNM1A foci will have to be evaluated in future studies, and for this reason we have simplified the Discussion section to simply highlight this issue.

2. In light of this, I still think the data in Figure 2A would benefit from some sort of quantitation to give a better idea of how many SNM1A foci are in proximity to 53BP1 vs how many (partially) overlap with 53BP1.

This is a good suggestion, and we have done this in new panel Fig. 2B, and associated text. The quantification is quite revealing. Both classes of SNM1A foci increase (53BP1 'overlapping' and 'proximal') while distant foci are reduced. As detailed above, future work will be required to understand the control of formation, and mechanistic impact of these different classes of foci.

3. "Also regarding Figure 3, is EGFP-SNM1A focus formation correlated to expression levels? If this is the case, the authors need to control for similar expression levels between mutants.

Rebuttal: "This is not an issue – we carefully assessed the baseline EGFP signal to ensure it was equal of each cell prior to striping to avoid this pitfall."

I meant the foci experiments in Figure 3 (not the laserline (striping) experiments in Figure 4). The authors need to show that all constructs were expressed at similar levels in order to be able to compare their focus forming properties. I find the sentence „we carefully assessed the baseline EGFP signal“ very dissatisfying. How was this assessed? There is no explanation or figure as far as I could tell.

We have updated the Materials and Methods (page 24-25) to more explicitly describe how cells were selected and analysed. Briefly, for each biological repeat, the laser power to ensure the maximum fluorescence in each channel was below saturation. Processing in image J using the macros script then normalised the fluorescence output using the Huang Dark auto threshold setting, as has been previously evaluated (see ref [https://doi.org/10.1016/0031-3203\(94\)E0043-K](https://doi.org/10.1016/0031-3203(94)E0043-K)). A (natural) distribution of whole-cell fluorescence of EGFP-SNM1A is still observed amongst the cell population. In response the reviewer's concerns, we have quantified this and display below as a RUG plot. As is apparent, while there is a range of

expression within the population, the cells evaluated were within the major population that represent relatively 'tight' population. Less than 2% of cell analysed had an area/mean intensity >2000.

Reviewer #3 (Remarks to the Author):

The revised version of Swift et al. manuscript is mostly similar to its original version, and I remain convinced that the data provided are insufficient to support the authors' claim that NSM1A promotes the repair of dirty ends. I therefore do not support publication in Nature Communications.

1- The authors did not answer my point #1 and did not provide foci accumulation analyses in a different cell line.

This has been done for the 293FT cells, and forms new Suppl. Fig. 4. The cells behave similarly to the U2OS versions.

2- The AsiSI experiment needs to be quantified. Stating that there is "a degree of proximity" between SNM1A and DSBs markers without any quantification is rather appalling. Since there is recruitment of SNM1A at clean DSBs, which is clearly in opposition to the authors' hypothesis, it is CRITICAL to show that 1) cell survival is unchanged and 2) 53BP1 and γ H2AX foci do not accumulate longer at AsiSI DSBs in SNM1A KO cells compared to WT cells – basically perform the analyses done in Figures 1 A-H but using AsiSI as inducer of DSBs.

We have scored these, shown in new panel Fig. 2D. This is quite revealing since while EGFP1-SNM1A is recruited some of the AsiSI DSBs, this recruitment tends not to be proximal or overlapping of γ H2AX. The fact that SNM1A is recruited to some 'clean' DSBs is not in opposition to our hypothesis. Some role in the repair of clean DSBs is perfectly possible and interesting in its own right. The key point is that SNM1A appears particularly important -- phenotypically -- at complex DSBs. There is no necessary conflict, both things can be true.

In respect of the additional experiments proposed: 1. Cell survival after AsiSI DSB induction; and 2. The accumulation/persistence of γ H2AX and 53BP1 foci; this cannot be done. A very comprehensive review by Gaëlle Legube and colleagues (the laboratory where the AsiSI system was developed) very nicely frames the limitation of these enzyme-based DSB induction systems (doi: 10.3389/fmolb.2020.00024). In essence, the enzyme doesn't cut all sites synchronously, but will continue to cut sites at various places in the genome over a long period, once expressed. Moreover, after faithful repair, the DSB site is nearly always cut again, iteratively, meaning that ultimately the cells will die of multiple rounds of DSB induction precluding this type of analysis. To quote from this review: 'for the above-mentioned I-SceI based systems: (i) DSB production is not immediate nor synchronized in the cell population and (ii) accurately repaired DSB can be re-cleaved. Hence while being powerful to analyze the spatial distribution of repair protein and chromatin changes around DSBs, they preclude a fine temporal resolution of these events.'

3- I asked for a simple quantification for Figure 3 and even that was not done. The manuscript is hugely lacking rigor regarding co-localization analyses.

This now done on new Supplemental Figure 10. We are grateful for the suggestion, as this analysis reveals some details of the localisation of EGFP-SNM1A after zeocin damage that are not apparent by scoring total foci: notably that the PIP box mutant forms of the protein show a much-diminished capacity to colocalise with 53BP1 and γ H2AX, which is consistent with the pivotal role for the PIP box in damage recruitment that emerged from our laser stripping experiments.

4- My point #4 was not addressed either.

As we mentioned previously, we acknowledge that this is a good suggestion. We find, however, that the treatment of these reporter plasmids reduces the (already) very low efficiency of repair even in WT cell in the assays precluding any useful conclusions to be drawn. Please see below for the data:

REVIEWERS' COMMENTS

Reviewer #2 (Remarks to the Author):

The authors have in my view made satisfactory revisions to the manuscript and I now endorse its acceptance for publication.

Reviewer #3 (Remarks to the Author):

The authors have answered most of my concern, and I do understand that some suggested experiments are not feasible. I do believe that at this point the manuscript is ready for publication in Nature Communications.

REVIEWERS' COMMENTS

Reviewer #2 (Remarks to the Author):

The authors have in my view made satisfactory revisions to the manuscript and I now endorse its acceptance for publication.

Reviewer #3 (Remarks to the Author):

The authors have answered most of my concern, and I do understand that some suggested experiments are not feasible. I do believe that at this point the manuscript is ready for publication in Nature Communications.

Response: We were pleased that the reviewers both felt that the work was ready for publication.